# Hectometric-scale modelling of the mixed layer in an urban region evaluated with a dense LiDAR-ceilometer network

Russell H. Glazer[1], Sue Grimmond[1], Lewis Blunn[2], Daniel Fenner[3,4], Humphrey Lean[2], Andreas Christen[3], Will Morrison[3,5], Dana Looschelders[3], Jonathan K. P. Shonk[2]

[1] Department of Meteorology, University of Reading, Reading, United Kingdom
[2]MetOffice@Reading, University of Reading, Reading, United Kingdom
[3]Chair of Environmental Meteorology, University of Freiburg, Freiburg, Germany
[4]Chair of Climatology, Technische Universität Berlin, Berlin, Germany
[5]School of GeoSciences, University of Edinburgh, Edinburgh, United Kingdom

*Correspondence to*: Russell Glazer, r.h.glazer@reading.ac.uk

**Abstract.** With development in recent years of hectometric (O(100 m); hm) scale numerical weather prediction (NWP) models, there is a need for their evaluation with high spatio-temporal scale observations. Here we assess UK Met Office Unified Model (UM) simulations with grid-spacing down to 100 m using a dense network of observations obtained during the *urbisphere*-Berlin campaign. A network of 25 automatic lidars-ceilometers (ALCs) provide aerosol attenuated backscatter observations from which mixed-layer height (MLH) is determined. UM simulated aerosol on two days (18 April and 4 August 2022) is used to determine model MLH with a novel algorithm (MMLH). Evaluation of MMLH with ALCs is focused on the MLH (1) urban-rural variability, and (2) urban plume. MMLH is consistently able to reproduce the vertical extent of the mixed layer during late afternoon despite the two case-study days having different maxima. MMLH performance is better in the 100 m model domain compared to a 300 m configuration, which may be explained by the higher vertical resolution in the 100 m configuration. During the August case in which an extreme heat event occurred, a delayed MLH growth is seen in the morning and afternoon over the city compared to the rural surroundings in both the model and ALCs. Both days show a distinct influence of the city through the mixed layer, including a plume extending downwind of the city that is detectable in both the observations and model. The modelled urban plume has a deeper mixed layer compared to the rural surroundings (4 August: ~500 m; 18 April: ~200 m) for up to 15 km downwind of the city.

## 1. Introduction

With nearly 70 % of the world's population projected to be living in and around cities by 2050 (UN 2015), there is a need for quality forecasts of weather and climate-associated risks and hazards with fine spatial resolution for densely populated areas and their infrastructure (Baklanov et al. 2018; WMO 2019). With the vulnerability to heat stress and extreme rainfall projected to increase with anthropogenic warming in urban areas (Birkman et al. 2016; Gencer et al. 2018; IPCC 2023), an increased capacity to deliver improved forecasts is critical.

Surface-atmosphere interactions are mediated by the atmospheric boundary layer (ABL) or urban boundary layer (UBL) when above urban areas. Urban/non-urban differences in surface properties cause different energy-balance partitioning creating a distinct UBL. With strong surface heating the UBL may be deeper than its rural counterpart upwind of the city and extend downwind as a distinguishable urban plume (Lowry 1977; Oke et al. 2017). Given the importance of UBL height for applications such as air quality, aviation, and convection, numerous studies of its characteristics have been undertaken (e.g. Schäfer et al. 2010; Schäfer et al. 2012; Tang et al. 2016; Geiß et al. 2017; Salvador et al. 2020; Bélair et al. 2025).

Due to the wide variety of techniques for determining ABL height (Stull 1988; Seibert et al. 2000, Lammert and Bösenberg 2006, Dai et al. 2014), metrics can differ depending on the atmospheric processes and variables analysed. A thermal inversion can be measured with balloon releases (Cleugh and Grimmond 2001; Wulfmeyer and Behrendt 2021) and aircraft-mounted sensors (AMDAR, Drüe et al. 2010; Kotthaus and Grimmond 2018a). Ground-based atmospheric lidars can observe turbulence (Barlow et al. 2011; Bonin et al. 2018) and/or aerosol particles (Kotthaus et al. 2018; 2020; Ritter and Münkel 2021) to diagnose the depth of vertical mixing in the ABL.

Advances in computational power and efficiency are enabling finer grid-spacing for numerical weather prediction (NWP), with some modelling centres developing or routinely running hectometric (i.e. O(100 m); hm) NWP models (Leroyer et al. 2014; Heinze et al. 2017; Ronda et al. 2017; Joe et al. 2018; Hanley and Lean 2024). Although hm-modelling computational costs limit domains to regions of O(10-100 km), they are well-suited to cover the metropolitan regions of large cities. At the UK Met Office, development and evaluation of the Unified Model (UM) at hm-scales is expected to lead to first applications for Greater London, UK (Boutle et al. 2016; Lean et al. 2019; Sützl et al. 2021; Lean et al. 2022; Blunn et al. 2023; Hall et al. 2024; Hanley and Lean 2024).

Given the fine-scale surface heterogeneity of cities (Stewart and Oke 2012; Hertwig et al. 2025), hm-scale models should provide benefits from capturing intra-urban variability of urban form and density, radiative and thermal material properties, vegetation, and emissions. The scales represented in hm-models mean the partial resolution of turbulent processes in the boundary layer (Honnert et al. 2011; Wyngaard 2004), with the potential for improved momentum and heat-flux representation. Urban forecasting using hm-models have shown improvements over current kilometre-scale (1-10 km) models in terms of turbulent mixing of pollution in convective boundary layers (Blunn et al. 2023), as well as heat stress, urban heat island (UHI) intensity, near-surface winds, and convection (Ronda et al. 2017; Joe et al. 2018; Hanley and Lean 2024). Finer resolution is also beneficial for climate applications (Hall et al. 2024) and for other urban services impacted by weather, environment or hydrological conditions (WMO 2019; Grimmond et al. 2020).

While the benefits of hm-models for urban areas may be intuitive, their relatively short grid lengths make comprehensive evaluation challenging or impossible using conventional observation networks that are generally too coarse and are not resolving differences at urban scales. Large roughness elements within cities also make WMO/traditional weather stations unsuited for evaluating sub-mesoscale to microscale processes in urban areas (Masson et al. 2002, Grimmond et al. 2010a, Lipson et al. 2024). As a result, there is a long-standing recognition of the need for dense 4-D observations in urban

areas to study poorly understood urban-scale processes, evaluate state-of-the-art hm-models, and for data assimilation (Grimmond et al. 2010b; NRC 2012; Barlow 2017; Scherer et al. 2019; Lean and Theeuwes et al. 2024).

    To overcome such challenges, a dense and systematic network of sensors was deployed in the city and surrounding regions of Berlin, Germany, during 2021-2022 (*urbisphere*-Berlin, Fenner et al. 2024). The network design enables upwind and downwind effects of the city to be captured, facilitating intra-urban and urban-rural comparisons. Measurement of the

spatio-temporal evolution of the ABL was a key component of the campaign and for this purpose used 25 vertically staring automatic lidar-ceilometers (ALCs). ALC attenuated backscatter from aerosols can be used to diagnose the vertical extent of recent mixing (hereafter mixed layer height; MLH) (Barlow et al. 2011; Bohnenstengel et al. 2015; Kotthaus et al. 2018, Kotthaus and Grimmond 2018a,b; Lean et al. 2022). For comparison with NWP, Warren et al. (2018; 2020) derived fields of UM simulated aerosol backscatter, to be comparable with observed aerosol backscatter from ALC sensors. A similar approach

is used by Scarino et al. (2014) over Southern California, with the simulated aerosol backscatter fields used as input in a MLH detection algorithm. As in Lean et al. (2022), MLHs can also be derived directly from model aerosols fields.

    The *urbisphere*-Berlin campaign provides an unprecedented opportunity for comparison and evaluation of the characteristics of the urban mixed layer with current hm-scale NWP models. As development of hm-scale NWP models is relatively recent, there is a need for more evaluation studies of their capabilities. Fenner et al. (2024) used UM hm-scale

simulations for comparison with surface energy balance observations taken during the *urbisphere*-Berlin campaign and initial ABL analyses, however a detailed and comprehensive evaluation of mixed-layer characteristics was not the focus of their study.

    Previous works have noted marked differences between various metrics for ABL height such as turbulence-based 'mixing height', aerosol-based MLH, and thermodynamically derived ABL height (Barlow et al. 2011; Collaud Coen et al.

2014; Kotthaus et al. 2018; Lean et al. 2022; Kotthaus et al. 2023a). These differences are due to the different atmospheric variables sampled. For example, MLH can be derived from analysing a concentration profile (e.g. aerosol concentration, moisture) whilst mixing height derived by analysing the strength of vertical motion (e.g. turbulence; Barlow et al. 2011; Lean et al. 2022). Given the sources and sinks of a scalar are not identical, the concentration profiles of temperature, moisture, and aerosols will not be identical (e.g. Roth and Oke 1995, Beyrich 1997). To ensure consistency between the model and the ALC

observations an aerosol-based NWP diagnostic is needed. Hence, we develop a new algorithm, 'MURK mixed-layer height' (hereafter MMLH) to diagnose MLH from profiles of NWP aerosol output (Section 3). MMLH is developed for use with the UM's single species aerosol scheme (termed 'MURK'; Clark et al. 2008) and separates the diurnal cycle of MLH into distinct periods similarly to the CABAM algorithm of Kotthaus and Grimmond (2018a). While some past studies have compared MLHs from one site (Lean et al. 2022) or one flight path (Scarino et al. 2014) with modelled MLHs, the current study compares

systematic MLH observations from 25 ALC sites throughout a large urban region with hm-scale UM simulations, allowing for a detailed study of the spatio-temporal evolution of the mixed layer over the metropolitan region of a major city along diurnal time scales.

Here we evaluate MMLH performance using the ALC sensor network (Section 2.2) on two *urbisphere*-Berlin campaign days (Section 2.3). Firstly, we aim to evaluate the ability of the UM (Section 2.1) at hm-scale to produce the variability of observed urban-rural MLH (Section 4.1 and 4.2). Secondly, we seek to better understand the characteristics of the urban plume in terms of MLH (Section 4.3) and to what degree the model reproduces an urban plume that is also detected in the ALC observations.

## 2. Methods

### 2.1 Case-study days

The two case-study days selected have clear-sky conditions, ensuring dry-convective ABL conditions and facilitating the use of aerosol as a tracer for the mixed layer. Case one in spring, 18 April 2022 ('April case'), had scattered clouds in some locations in the afternoon. A colder airmass approached the Berlin region from the Baltic Sea from the northeast during the afternoon and evening (Fig. 1a), resulting in a northeasterly-dominated flow over the city. Case two, 4 August 2022 ('August case'), occurred near the climax of an intense western-central European heat wave (Copernicus Observer 2022). On 4 August, eastern Germany was located in a region of persistent warm southerly flow (Figure 1b) contributing to 2 m air temperatures that reached up to 37 $^\circ$C in Berlin (Fenner et al. 2024, Appendix A1).

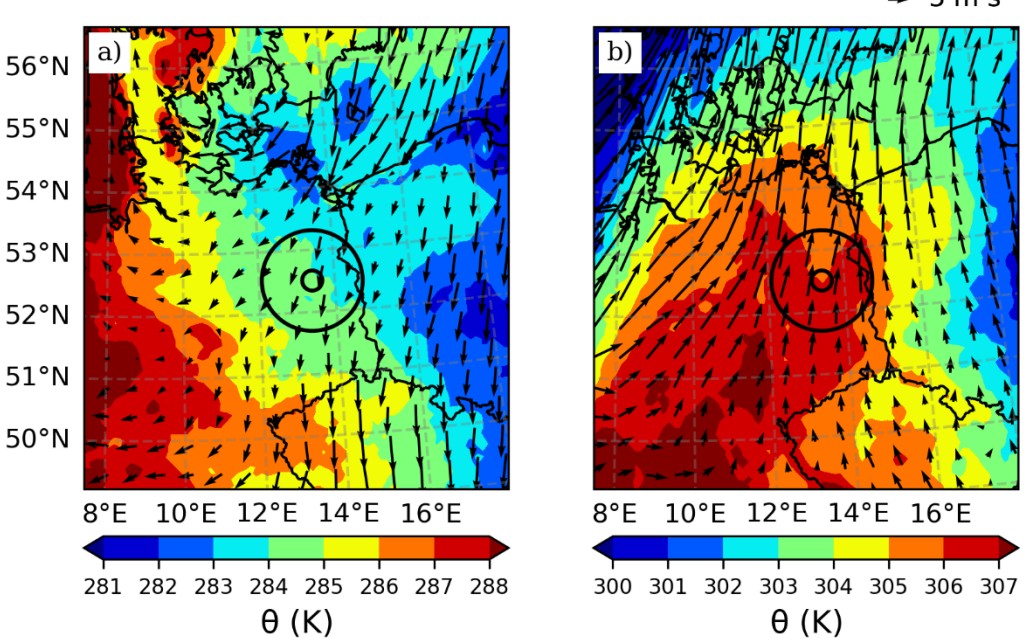

**Figure 1.** Global UM (10 km grid resolution) modelled 850 hPa potential temperature ($\theta$, shaded, scales differ) and wind vectors (scale at top right) hourly averaged with rings B and C (black circles) and country borders (black lines) shown at 14:00 UTC on (a) 18 April, and (b) 4 August 2022.

### 2.2 Unified Model (UM) configuration

We use the UK Met Office UM (version 13.0; Davies et al. 2005; Wood et al. 2014) one-way nested within the 10 km Global UM (Global Atmosphere v8.0 and Global Land v9.0 science configurations, updated from Walters et al. 2019). Three nested domains (Fig. 2a) are used with grid-spacings of 1.5 km, 333 m (referred to as 300 m model), and 100 m with 70, 70, and 140 vertical levels, respectively (Table 1). In the 70 vertical level grid-set, there are 16 levels in the lowest kilometre above the model surface (ams) with the first level at 5 m ams. In the 140-level configuration the first level is at 2 m ams, with 37 levels in the lowest kilometre. In the 300 m domain, the vertical spacing below 200 m is less than ~60 m increasing to ~120 m at 1 km, whereas in the 100 m domain it is ~20 and ~40 m respectively (Fig. S1). The three nested domains are centred over the central point of Berlin derived from urban form data in Fenner et al. (2024). The 300 m domain contains all *urbisphere*-Berlin observation sites, whilst the 100 m domain (Fig. 2b) extends 30 km from the city centre.

The simulations use the Regional Atmosphere and Land science configuration version 3.1 (RAL 3.1; Bush et al. 2024) which encompasses the physics schemes in the model such as radiation, and microphysics. The turbulence scheme used is 'scale-aware' (Boutle et al. 2014) which practically means a conventional 1D non-local turbulence scheme (Lock et al. 2000) is used when the ABL is shallow and the length scales for turbulence are small compared to the grid length (i.e. not resolved well). When the ABL is deeper, and turbulence has a larger length scale (i.e. more likely resolved by a hm-scale model) a local 3D Smagorinsky (1963) scheme has more weighting. Thus, weighting depends on the ABL depth and horizontal grid-spacing. This implies that as grid-spacing decreases, the turbulence in the model increasingly comes more from the 3D Smagorinsky scheme and less from the conventional parameterised scheme. This is important when considering the difference between the 100 and 300 m simulations in the representation of turbulence in the ABL, as we might expect the 100 m model to begin using the 3D scheme earlier in the day compared to the 300 m model.

The JULES (Joint UK Land Environment Simulator v7.0; Best et al. 2011, Clark et al. 2011, Walters et al. 2019) land-surface model with the MORUSES option is used for built land-cover fractions (Porson et al. 2010a, b, Bohnenstengel et al. 2011) split between roof and street canyon tiles. Land-cover fractions are derived from European Space Agency (ESA) Climate Change Initiative (CCI) version 1 data (ESA 2017). Anthropogenic heat flux varies monthly based on a UK-based mean value and built land-cover fraction in each grid cell (Hertwig et al. 2020).

Within the UM a single-species aerosol scheme is included which is described in Clark et al. (2008). Aerosol species are grouped into a single-species, referred to as 'MURK'. The scheme defines the concentration of aerosol as a prognostic quantity that is advected via a semi-Lagrangian tracer advection scheme and mixed consistent with other scalar quantities in the model (e.g. water vapour). Aerosol sinks are accounted for via precipitation and non-emissions sources are treated via a conversion factor. Surface emissions are provided by the European Monitoring and Evaluation Programme (EMEP/CEIP 2023) at 0.1° resolution. A sinusoidal diurnal variation is applied to the surface emissions with the peak occurring at midday. In the model, emissions are vertically distributed near the surface depending on type of source. Most sources are from area emissions and are evenly distributed in model levels between the model surface and a height of 150 m. For large point sources, emissions are spread evenly over model levels within a minimum and maximum plume height for the source. When the plume height is unknown, the minimum is set to 150 m and the maximum to 365 m.

For the two case study days the UM is run for 36 hours, from 12:00 UTC (local time is UTC+2) on the day prior to each study day. The first 12 h are treated as spin-up. This is found to be important for the MURK aerosol as it needs to be transported from the surface emissions across the domains.

For evaluation of the model output to observational data at specific sites, we use the nearest UM grid cell to the observation site. All times referred to in this paper are UTC.

Hall et al. (2024) previously noted biases in the downscaling of soil moisture and soil moisture stress from the Global UM resulting in sharp gradients and a wet bias in urban areas. As this bias appears amplified during the heatwave/drought conditions of 4 August, we replaced the urban soil moisture state with the surrounding rural area average state (see Section S2).

**Table 1.** Model domains configurations and physics schemes used for all simulations

| Domain | 1.5 km | 300 m | 100 m |
|---|---|---|---|
| # vertical levels | 70 | 70 | 140 |
| # vertical levels in lowest km | 16 | 16 | 37 |
| # grid pts. [x,y] | 334 x 334 | 650 x 730 | 702 x 702 |
| Turbulence | Boutle et al. (2014) | | |
| Physics schemes | RAL3.1 (Bush et al. 2024) | | |

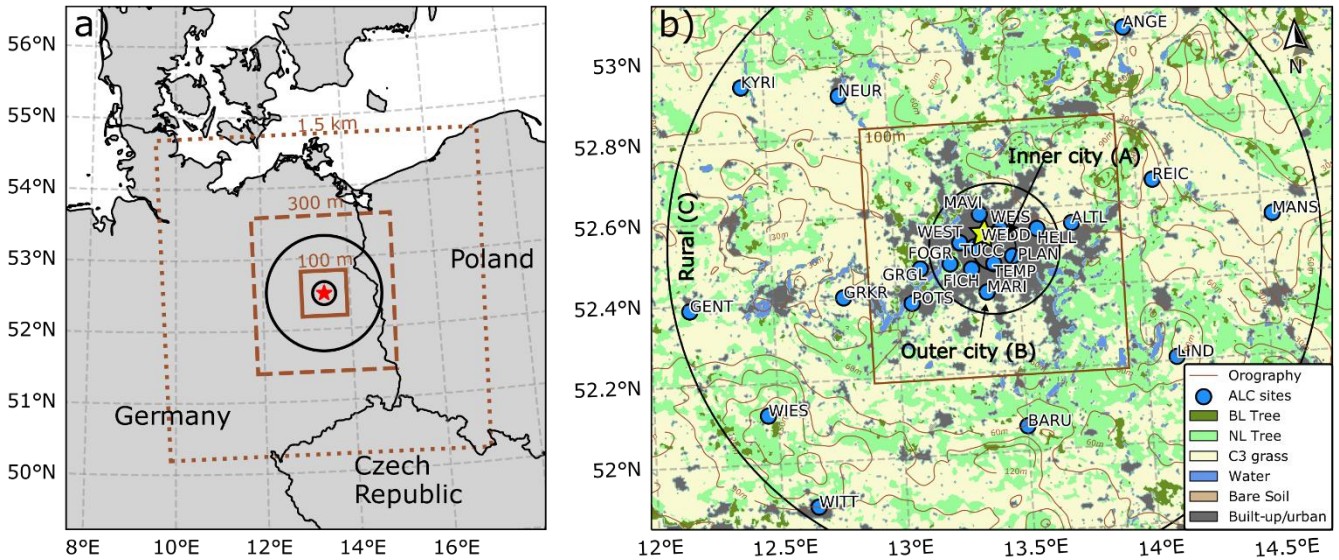

**Figure 2.** Berlin study area with (a) the extent of the three nested model domain (brown, 100 m, 300 m, and 1.5 km) and   rings B and C (black circles) around Berlin (red star), and (b) location of the 25 ALC sites (blue, with  WEDD [Wedding] - yellow star) with the 300 m domain model dominant land-cover (colour, BL = Broadleaf, NL = Needleleaf) and orography (30 m contours), plus the A to C rings (black circles) at 6, 18 and 90 km radius from Berlin centre, locating Inner city, Outer city and rural areas, respectively (Fenner et al. 2024) and extent of 100 m model domain (brown box).

**2.3 *urbisphere*-Berlin campaign ALC data**

Between autumn 2021 and autumn 2022 the *urbisphere*-Berlin campaign deployed 25 ALC sensors (Table S1), including 11 within Berlin's city limits and 14 within the surrounding rural area but within 90 km of Berlin's centre (Figure 2b). The three ALC models used are Lufft CHM15k (13), Vaisala CL31 (7), and Vaisala CL61 (5), with not all operational at all times [Fenner et al. (2024, SM2)]. As some ALCs were deployed on roofs of tall buildings, all MLH measurements are adjusted to metres above ground level (m agl). On the case-study days (Section 2.3) the number of operating ALCs were 24 (18 April 2022) and 21 (4 August 2022). To facilitate intra-urban and urban-rural analyses, Fenner et al. (2024) define three spatial extents/rings - inner (A, radius 6 km) and outer (B, 18 km) city, and rural (C, 90 km). The 100 m model domain covers all of *urbisphere*-Berlin inner (A) and outer (B) city rings, but not the entire rural (C) ring (Fig. 2b). The terrain surrounding Berlin is generally flat with low hills (~90 m) to the northeast and southwest. Vegetation in the rural (ring C) areas are dominated by either croplands or needle leaf forests.

The ALC-observed attenuated backscatter profiles are analysed with sensor specific algorithms to derive the observed MLH (hereafter $Z_O$). CABAM (Kotthaus and Grimmond 2018a), which was designed for low signal-to-noise ratio (SNR) ALC is used for the Vaisala CL31s, and STRATfinder (Kotthaus et al. 2020; Kotthaus et al. 2023b), designed for high-SNR ALC, for the Vaisala CL61s and Lufft CHM15k devices. Kotthaus et al. (2020) compared these two ALC algorithms systematically and found very good agreement in MLH in diverse ABL conditions. However, the largest discrepancies between the two can occur in early morning and evening when detecting shallow and weak layers that may be more sensitive to the difference in SNR and optical overlap. The data processing in STRATfinder yield a constant vertical spacing of 30 m up to ~15km height. For CABAM the vertical spacing is maintained for the original CL31 range gates which is 10 m up to ~7.5 km height. The $Z_O$ data are aggregated to 15-min means. Fenner et al. (2024) provides details about sensor location and data processing chains.

During stable or nocturnal stratification periods the ABL can be shallow (< 200 m). The Lufft CHM15k optical overlap is incomplete at these heights, making data unavailable in the early morning and after sunset for these devices, whereas the CL31 and CL61 sensors reach sufficient optical overlap at lower heights and with suitable overlap correction built in (Kotthaus et al. 2020), hence data availability is better for these periods. At sunset, with developing nocturnal stratification, weak aerosol backscatter gradients make confidence in deriving $Z_O$ low.

**3. MURK Mixed-layer Height (MMLH) algorithm**

To compare the observed MLH ($Z_O$) with the model we post-process the UM MURK aerosol output (15-min mean aerosol content (*A*), 3-D field). Given an understanding of the mixed-layer diurnal cycle, the MMLH algorithm splits the day into four periods (Morning, Growth, Afternoon, and Evening, colour Figure 3) in order to diagnose the model MLH (hereafter $Z_M$).

The vertical *A* profile is initially temporally averaged over five model output intervals (i.e. 1.25 h) with central weighting ([0.125,0.25,0.25,0.25,0.125]) to retain continuity between timesteps when abrupt changes in the profile may

significantly affect MLH diagnosis. Given the study region and days of year analysed (Section 2.3), the algorithm analysis begins at 00:00 UTC (i.e. continuity from prior day not needed). In the morning period, the largest vertical gradient ($\nabla$) in $A$ below 700 m is searched for, with $Z_M$ assigned to the model level directly above $\nabla$. Vertical gradients of $A$ are calculated using a least squares regression on three model vertical levels above and below a given level.

When the mixed layer has begun growing, at time $t_{grow}$, the Growth period (black, Fig. 3) begins and $Z_M$ is found by searching from the lowest model level (z=1) vertically upwards (i.e. increasing z) by comparing $A_z$ relative to $A_{z=1}$ with $Z_M$ found at the first level at which $A_z <$ 10 % of $A_{z=1}$. This threshold is derived from empirical testing with model data, informed by comparison to $Z_O$.

After sufficient growth has occurred, at a time $t_{aft}$, the Afternoon period (green, Fig. 3) begins and $Z_M$ is determined by searching from higher to lower model levels, to find the first model level at which $A_z >$ 10 % $A_{z=1}$ (i.e., same threshold as Growth period). The Evening period (blue, Fig. 3) begins at sunset ($t_{SS}$). In Evening, $Z_M$ is found using the same criteria as during the Morning period.

To determine the transition times ($t_{grow}$, $t_{aft}$), a first iteration of the algorithm estimates the daily minimum ($Z_{M,min}$) and maximum ($Z_{M,max}$) MLH assuming they occur between sunrise ($t_{SR}$) and solar noon ($t_{SN}$) for $Z_{M,min}$, and between $t_{SN}$ and sunset ($t_{SS}$) for $Z_{M,max}$. The difference between these gives an estimate of total MLH growth ($\Delta Z_{MLH} = Z_{M,max} - Z_{M,min}$) for the day. Fractions ($F$) of this daily growth are used to determine when $t_{grow}$ and $t_{aft}$ occur, based on the first time a threshold is met:

$$Z_M > [\Delta Z_{MLH} \cdot F + Z_{M,min}], \tag{1}$$

with F=0.05 and F =0.5, for $t_{grow}$ and $t_{aft}$, respectively. Values for $F$ are informed by ALC data obtained from London (Kotthaus and Grimmond 2018b; Hertwig et al. 2021). $t_{grow}$ and $t_{aft}$ are determined locally at each grid point and thus have spatial variability across the model domain. By letting the transition times be a function of the total growth of the mixed layer ($\Delta Z_{MLH}$) for a given day, the algorithm is more widely applicable to a range of MLHs. Following the determination of the transition times, iteration of MMLH can proceed throughout the day yielding 3-D (x,y,t) fields of $Z_M$.

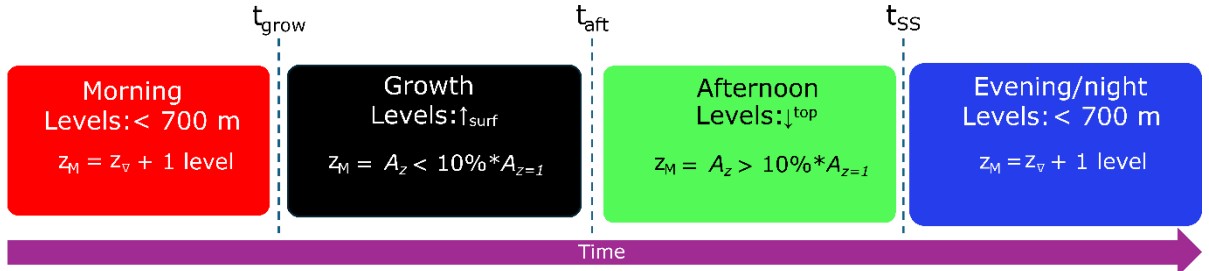

**Figure 3.** Schematic scheme of the MURK mixed layer height (MMLH) algorithm. MMLH is applied on a daily basis from the morning (here 00:00 UTC) for each model grid point to obtain $Z_M$. The Morning (red) and Evening (blue) periods require the height of the maximum gradient ($Z_\nabla$) in MURK aerosol ($A_Z$), while Growth (black) and Afternoon (green) periods depend on exceedance of a 10% MURK aerosol threshold with respect to the lowest model level aerosol concentration ($A_{Z=1}$). The Growth and Afternoon periods begin at onset times $t_{grow}$, and $t_{aft}$, respectively, following equation (1). Evening/night begins at sunset ($t_{SS}$).

## 4. Results and discussion

### 4.1 Site-based evaluation of MMLH

First, we evaluate $Z_M$ determined using the MMLH algorithm (Section 3) with $Z_O$ derived from ALC observations in Berlin and its surroundings (Section 2.2). Initially, we focus on an inner-city site (WEDD; Figure 2b, star) north of the city centre where a Vaisala CL61 ALC was deployed.

The time series on both days show consistent agreement between $Z_O$ and $Z_M$ in the morning and when the mixed layer begins to grow, especially for the August case (Fig. 4). In the morning of the August case, a residual layer is present in both the model (Fig. 4b, d) and the ALC observations (Fig. S5) which appears to cause $Z_M$ to be too high during the growth period in the 300 m domain (Fig. 4b). After sunset ($t_{SS}$), both the MMLH and STRATfinder algorithms begin looking for the top of a nocturnal stable layer which is well captured by MMLH on both case days. At around 21:30 in the August case a relatively short but sharp increase in MLH is seen in both $Z_M$ and $Z_O$.

Comparing the two model configurations, there is an improved ability in the 100 m model (Fig. 4c, d) in distinguishing the mixed layer from a residual layer during the Growth period, compared to the 300 m (Fig. 4a, b). This is possibly the result of increased vertical resolution, which allows for sharper vertical gradients in aerosol, and/or increased horizontal resolution leading to a better representation of turbulence.

For comparison, Figure 4 also shows a thermodynamic ABL height metric ('BLD diagnostic', grey, Fig. 4) that is available as diagnostic output from the model. During unstable conditions (e.g. afternoon) the BLD diagnostic uses a parcel method which is defined as the height to which an air parcel can rise dry adiabatically from the surface via convection up to its intersection with the environmental temperature profile (Holzworth 1964). In stable conditions it is the first height above the model surface where the gradient Richardson number is supercritical (>0.25). In the late afternoon and evening, a marked discrepancy is evident between BLD and both $Z_O$ and $Z_M$, especially for the August case. This is consistent with expectations from thermodynamic and aerosol-based observations (Collaud Coen et al. 2014; Kotthaus and Grimmond 2018a). However, it is notable that this is demonstrated in the hm-model simulations themselves, and the ALC observations support this. Possible explanation for this is seen in the continued strong vertical velocities (cf. April case Fig. S7a,c) in the late afternoon above BLD (Fig. S7b,d), which imply that additional vertical mixing is possible above a weak thermal inversion (Fig. S6b,d). Additionally, Lean et al. (2022) identified wave features above the mixed layer which may be contributing to what is seen in Fig. S7b,d.

Next, we evaluate MMLH at each ALC location (Figure 5), with ALC sites outside the 100 m domain (Table S1) using 300 m model MMLH values. On both days there is a tendency for $Z_M$ to overestimate MLH in the Afternoon compared to $Z_O$ (green, Fig. 5), especially at the highest MLHs in the August case. However, in both the Morning and Evening periods on 18 April $Z_M$ has a negative bias (red and blue, Fig. 5a,c).

Despite the large difference in maximum MLH between the two cases, MMLH has consistent behaviour both in terms of the separation of periods and in comparison to the ALCs. During the Morning and Evening periods of the August case (Fig

5d) there is consistent agreement when MLHs are below ~200 m. In the April case (Fig. 5c) an underestimation of MLHs <200

265    m occurs, which is related to weak aerosol gradients causing $Z_M$ to be diagnosed close to the model surface (not shown). Given the difference in maximum MLH between April and August, the normalized mean absolute error (nMAE, Appendix A1) is used to quantify the deviation of the model from the ALCs in each ring. On both days nMAE becomes larger with distance from the city centre (ring A to C, urban to rural areas) (Fig. 5a,b). The overall mean absolute differences between $Z_O$ and $Z_M$ (18 April (170 m); 4 August (289 m)) are an order of magnitude higher than that found in an intercomparison of CL61 ALCs

(Looschelders et al. 2025), suggesting differences between $Z_O$ and $Z_M$ can largely be attributed to model error.

To assess the performance of MMLH as a function of time we use hit rate thresholds of 10 % ($HR_{10}$) and 50 % ($HR_{50}$) (Appendix A2) in Figure 6. Performance is generally better in the afternoon (12:00-18:00) on both days with HR >80 %, even for the more stringent $HR_{10}$ threshold, in the 100 m domain. Outside of the afternoon, there is a maxima in HR in the morning in both cases: at around 06:00 ($HR_{50}$~80 %) on 18 April and between 06:00-09:00 ($HR_{50}$=100 %) on 4 August. This may relate

to strengthening aerosol gradients before the mixed layer begins to rapidly grow. When HR is normalised by the hit window (Fig. S8; Appendix A2) the performance during this morning period stands out. Additionally, there is increasing hit rate into the evening (after SS) on both days which could be related to a strengthening of the nocturnal stable layer as the evening progresses. Interestingly, the HR for the 300 and 100 m domains are similar for the April case (Fig. 6a,c) whereas for the August case the 100 m domain has higher HRs compared to the 300 (Fig. 6b,d).

Note that there are generally more ALC sites with data in the April case compared to the August case (black line, Fig. 6). As noted in Section 2.2, there is a drop in ALC data availability at $t_{SS}$ in both cases that is associated with weaker backscatter gradients making MLH diagnosis challenging.

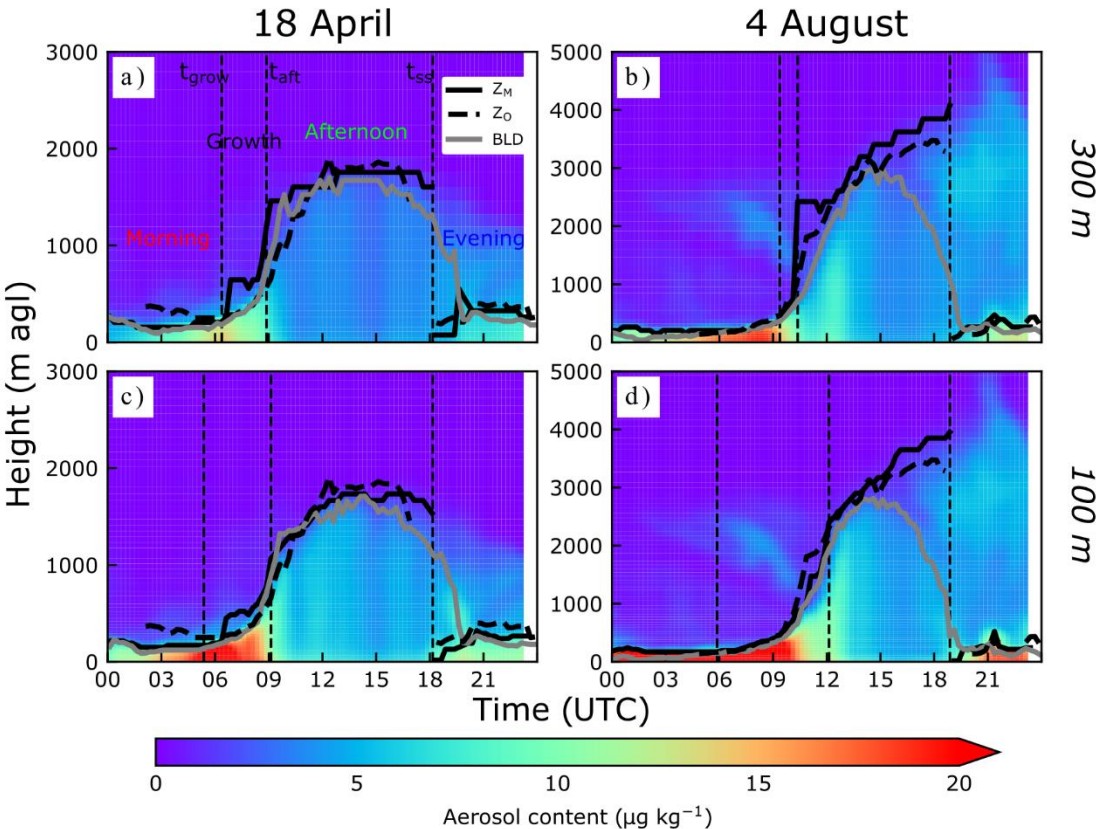

**Figure 4.** UM MURK aerosol content (shaded, µg kg-1) is used to derive mixed-layer height (MLH) at site WEDD (Fig. 2b, yellow star) with the MMLH algorithm ($Z_M$, Section 3, solid black line), ALC observations are used to obtain MLH using STRATfinder ($Z_O$ thick dashed black line), and UM boundary layer diagnostic (BLD; grey solid line) on (a, c) 18 April and (b, d) 4 August 2022 for (a, b) 300 m and (c, d) 100 m model domains. MURK aerosol is temporally smoothed (Section 3). In (a) the MMLH period is labelled as in Fig. 3, with each transition ($t_{grow}$, $t_{aft}$, $t_{SS}$) denoted by a thin dashed black line. Note different y axis scales for the two days.

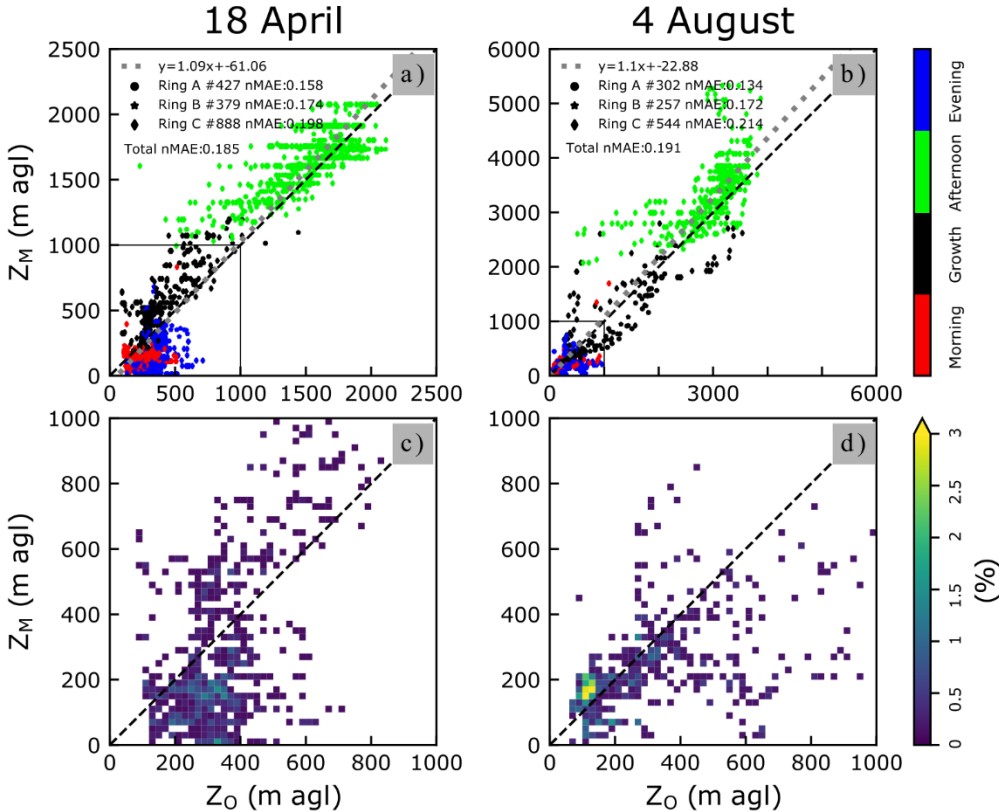

**Figure 5.** Comparison between MMLH ($Z_M$) and ALC MLH ($Z_O$) at all *urbisphere*-Berlin ALC sites (Fig. 2b) on (a) 18 April, and (b) 4 August 2022 (1:1 dashed line), with points coloured by MMLH algorithm period (Fig. 3), and linear regression shown as a dotted grey line (equation at top left). The number of data points per ring (#), and normalized mean absolute error (nMAE; Appendix A1) are shown in the top left. In (c, d) the probability density (%) using 20 m MLH bins for the lowest 1000 m is shown (extent indicated by black box in a, b).

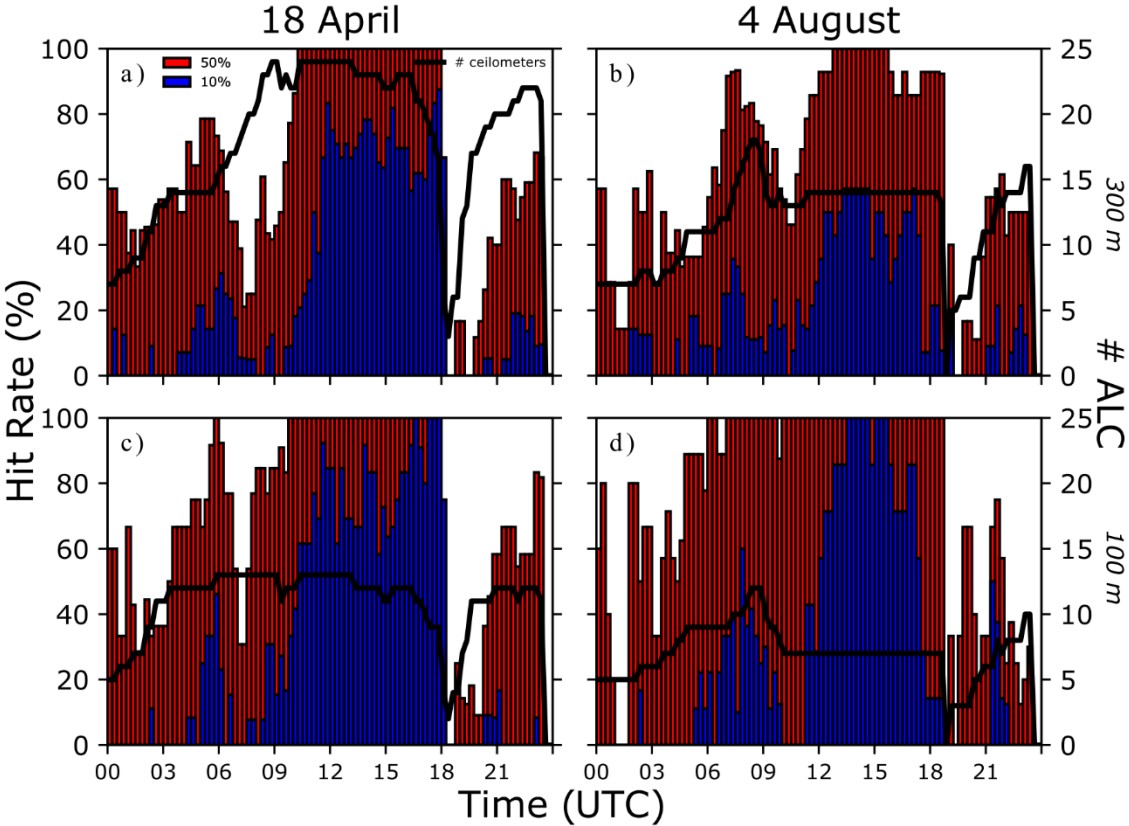

**Figure 6.** Model performance assessed using hit rate (Appendix A2) within 50 % (red) and 10 % (blue) of observed values by model resolution (a-b) 300 m, (c-d) 100 m, for (a, c) 18 April and (b, d) 4 August 2022. Number of ALCs giving data (black line) decreases near sunset because of ALC algorithm (Section 2.2).

## 4.2 Spatial Characteristics of MMLH

In addition to the MMLH performance relative to the individual ALCs, we can also exploit the full 2-D spatial fields of $Z_M$ to investigate the spatial characteristics of the mixed layer at both regional and neighbourhood scales, in conjunction with the network of 25 ALC sites.

First, we consider the April case at four times (03:00, 09:00, 15:00, and 21:00 UTC [05:00, 11:00, 17:00, 23:00 local time]) for both the 300 m (Fig. 7a-d) and 100 m (Fig. 7e-h) domains. The model bias error is shown at each ALC site. Table 2 also provides a summary comparing various metrics between $Z_M$ and $Z_O$, separating stations by spatial groups of rings A, B, and C, and whether stations were upwind or downwind of the city centre (excluding city sites).

Although at 03:00 (Fig. 7a,e, before SR) the mixed layer is generally characterised by nocturnal stable conditions, the influence of the urban surfaces is evident in the MLH, especially in the 100 m domain (Fig. 7e). The mean flow on this day is from the northwest, and higher MLHs occur roughly downwind of the urbanized land surfaces in the model. In Table 2 between 03:00-05:00 there is a marked difference going from A to C rings in $Z_M$ with a higher MLH in ring A compared to C by as

much as ~100 m (100 m domain). In the morning, the model appears to show an urban bulge in MLH. However, this does not seem to extend outside the city. In contrast, ALC sites ($Z_O$) show very little variability spatially in the morning.

At 09:00 the mixed layer is actively growing with increasing sensible heat fluxes, however this growth is not uniform and $Z_M$ is lower downwind of the city in the 100 m domain (Fig. 7f). Compared to $Z_O$ there is a general overestimation by $Z_M$, however, this is reduced in the 100 m domain. This could be the result of a faster growth rate in the model on the 300 m domain compared 100 m (Table 2).

In late afternoon (15:00, Fig. 7c, g) $Z_M$ is more uniform across the region, however, the ALC observations indicate a possible urban plume with higher MLH in the downwind areas (cf. $Z_M$). In the afternoon (13:00-15:00, Table 2) the $Z_O$ downwind MLH is higher by ~300 m compared to upwind, with the model also showing this difference. In the 100 m domain (Fig. 7g) $Z_M$ is too shallow downwind of the city, whereas upwind it is too deep (cf. $Z_O$). Ignoring wind direction, $Z_M$ in in the 300 m domain for rural sites (Fig. 7c) is too deep (cf. $Z_O$) at this time (Table 2b).

At night (21:00, Fig. 7d, h) a downwind effect of the city is apparent in $Z_M$, consistent with Fenner et al. (2024). Over urban surfaces downwind of the city centre the MLH is deeper compared to non-urban land surfaces around the city. The observations indicate the 300 m domain $Z_M$ is generally underestimated, especially downwind of the city (Fig. 7h, Table 2). That is, an urban plume is observed by the ALCs and is deeper than in the model at this time. Similar statistical significance between rings in $Z_M$ is indicated during evening and morning (Table 2). However, for $Z_O$ this only occurs in the evening. Thus, while the model tends to show an urban influence on MLH in both morning and evening, the observations only support this in the evening.

Second, for the August heatwave case, at 03:00 the spatial variability in rings A to C is similar to the April case with a higher MLH over urban areas in $Z_M$ in both domains (Fig. 8e, Table 3). By 09:00 the mixed layer is growing rapidly in most rural areas. However, near and over the city the growth is delayed (Fig. 8b,f), suggesting the soil moisture is still too large (Section 2.3), resulting in too much energy going to the latent heat flux (Fenner et al. 2024). The growth rates (Table 3) indicate steeper growth in ring C in $Z_O$ and $Z_M$ (100 m) possibly indicating that the mixed layer is indeed slow to grow in the urban areas. At 15:00, $Z_M$ is generally too high in the 300 m domain (Fig. 8c), but in the 100 m domain this model deviation from the ALCs is improved (Fig. 8g; Table 3).

With the mean flow generally from the south on this day, any urban plume would be expected to the north of the city. At 21:00 $Z_M$ (Fig. 8d, h) is deeper north and east of the inner-city (ring A) area and to a lesser degree following the urban land cover in the outer city urban extent (ring B). The $Z_O$ data suggest a shallower MLH over the city compared to $Z_M$ (Fig. 8h). For Downwind sites both $Z_O$ and $Z_M$ (300 m) are deeper than upwind (Table 3), indicating an urban influence on the mixed layer downwind of the city. However, with fewer sensors available in the plume direction (cf. April case) it is harder to clearly evaluate this. At the one sensor site north of ring A (Fig. 8h) $Z_M$ is ~500 m too low, while for all surrounding sites there is little difference or $Z_M$ is too high. Between 20:00-22:00 (Table 3) there are significant differences found between upwind and downwind for $Z_M$ (300 m).

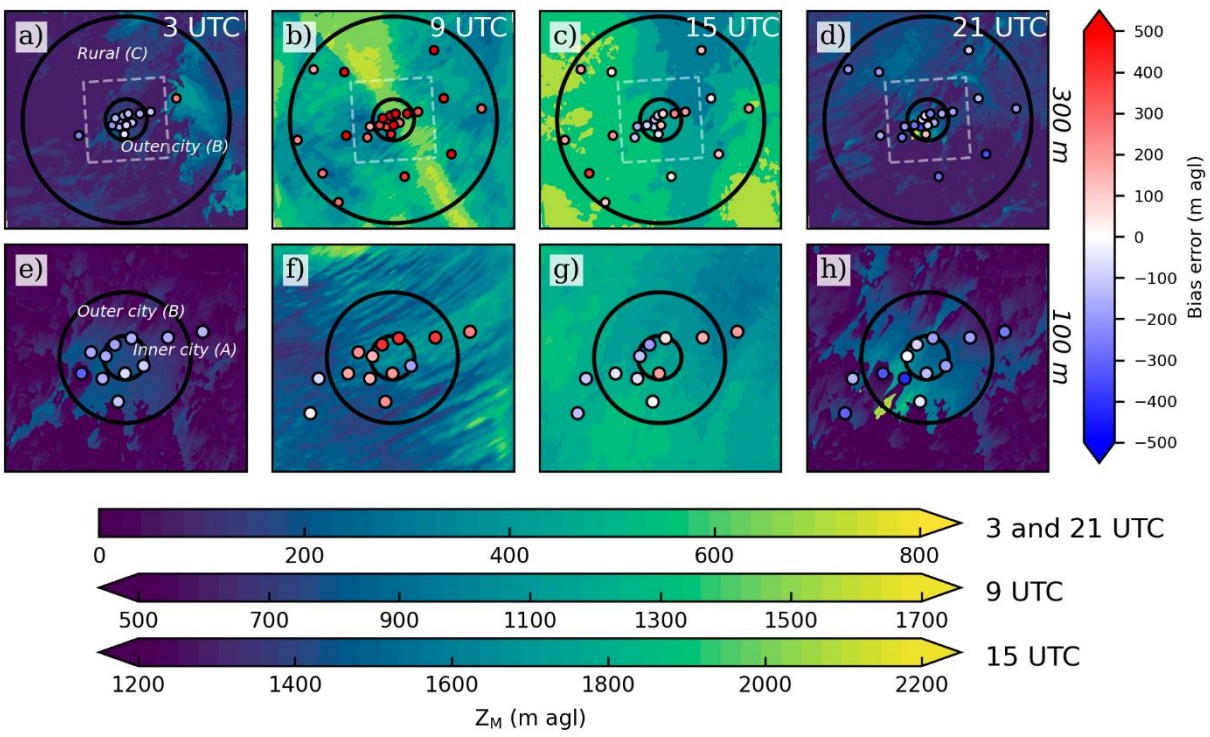

**Figure 7.** 18 April 2022 MMLH ($Z_M$) (shaded, time specific colour bars) at (a, e) 03:00, (b, f) 09:00, (c, g) 15:00, and (d, h) 21:00 UTC for (a-d) 100 m and (e-h) 300 m domains, with ALC sites (circle) $Z_M$ - $Z_O$ bias error (colour), extent of 100 m domain (white dashed box), and rings (black) A (6 km radius), B (18 km), and/or C (90 km) shown.

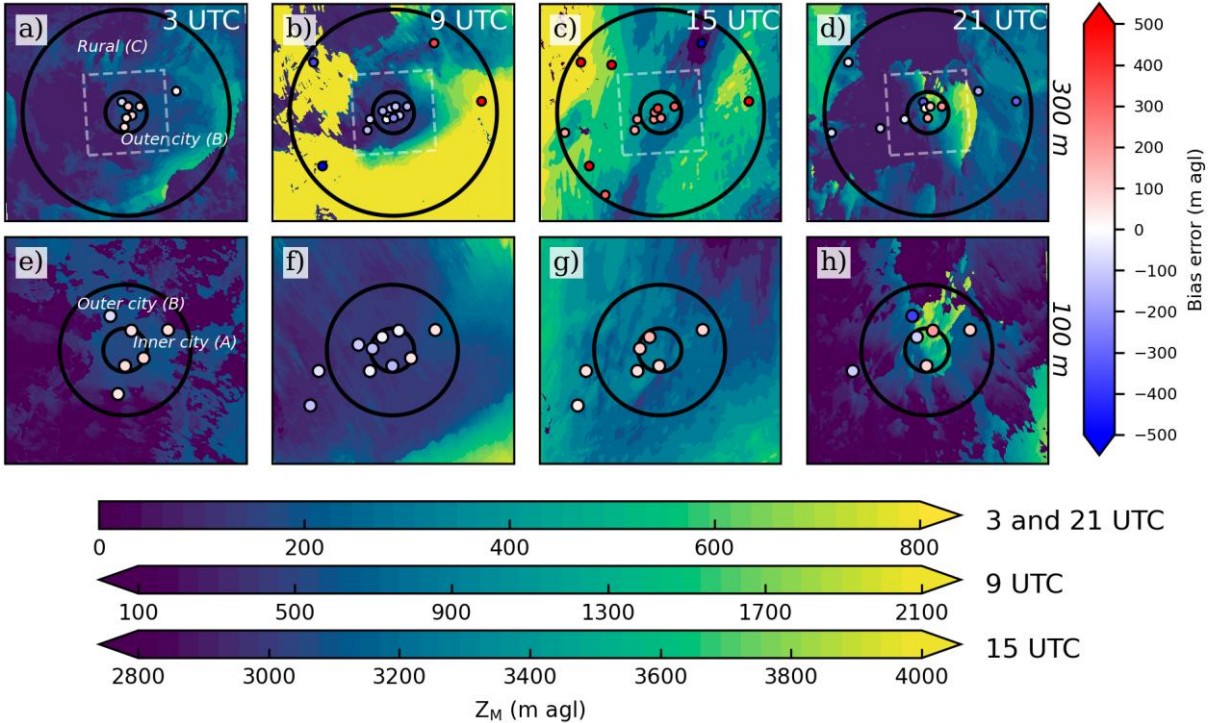

**Figure 8.** As Figure 7, but for 4 August 2022.

**Table 2.** Comparison between $Z_O$ and $Z_M$ (100 and 300 m domains) for 18 April using various spatial groupings of sites: Rings (A,B,C), and rural sites (Ring C) split into upwind (Up) / downwind (Do) / Other (Ot), showing the mean and (number of sites analysed) in each column, and the mean bias error (MBE, $Z_M$-$Z_O$) for three time periods (03:00-05:00, 15:00-17:00, 20:00-22:00). The Growth rate (Kotthaus and Grimmond, 2018b) and Maximum are also given. Significant difference testing uses Kruskal-Wallis (1952) with subsequent post hoc testing following Dunn (1964), and Holm (1979). Rings A and B are classified as 'City' for the purpose of significance testing with the Up, Do, and Ot groups.

| | A | B | C | Upwind | Downwind | Other | Significant Difference |
|---|---|---|---|---|---|---|---|
| **03:00-05:00** | | | | | | | |
| $Z_o$ | 242 (5) | 255 (5) | 237 (5) | 123 (2) | 312 (3) | (0) | |
| $Z_m$ 300 m | 175 (5) | 160 (5) | 155 (14) | 254 (5) | 97 (7) | 112 (2) | A-C; B-C; Up-City; Do-City; Ot-City; Up-Ot; Do-Up |
| $Z_m$ 100 m | 170 (5) | 133 (5) | 68 (3) | 42 (1) | 82 (2) | (0) | A-C; B-C |
| MBE 300 m | -59 (5) | -85 (5) | -44 (5) | 156 (2) | -178 (3) | (0) | |
| MBE 100 m | -75 (5) | -117 (5) | -99 (3) | -99 (1) | -99 (2) | (0) | |
| **Growth rate (m h⁻¹)** | | | | | | | |
| $Z_o$ | 307 (5) | 262 (5) | 285 (14) | 282 (4) | 279 (7) | 301 (3) | |
| $Z_m$ 300 m | 306 (5) | 320 (5) | 302 (14) | 207 (4) | 339 (7) | 340 (3) | |
| $Z_m$ 100 m | 246 (5) | 279 (5) | 246 (3) | 219 (1) | 259 (2) | (0) | |
| **13:00-15:00** | | | | | | | |
| $Z_o$ | 1730 (5) | 1734 (5) | 1737 (14) | 1506 (3) | 1824 (6) | 1770(5) | Do-Ot; Do-City; Do-Up |
| $Z_m$ 300 m | 1755 (5) | 1725 (5) | 1837 (14) | 1605 (3) | 1887 (6) | 1916(5) | A-C; B-C; Do-Ot; Do-Up; Do-City; Ot-City |
| $Z_m$ 100 m | 1733 (5) | 1734 (5) | 1712 (3) | 1600 (1) | 1768 (2) | (0) | |
| MBE 300 m | 18 (5) | -2 (5) | 98 (14) | 99 (3) | 52 (6) | 152 (5) | |
| MBE 100 m | 3 (5) | -8 (5) | -72 (3) | 127 (1) | -172 (2) | (0) | |
| **Maximum (m)** | | | | | | | |
| $Z_o$ | 1795 (5) | 1826 (5) | 1846 (14) | 1626 (3) | 1933 (6) | 1873(5) | Up-Do |
| $Z_m$ 300 m | 1755 (5) | 1755 (5) | 1870 (14) | 1605 (3) | 1913 (6) | 1977(5) | Up-Do; Up-Ot; Do-City; Ot-City |

| | A | B | C | Upwind | Downwind | Other | Significant Difference |
|---|---|---|---|---|---|---|---|
| $Z_m$ 100 m | 1761 (5) | 1775 (5) | 1734 (3) | 1666 (1) | 1768 (2) | (0) | |
| **20:00-22:00** | | | | | | | |
| $Z_o$ | 382 (5) | 312 (4) | 321 (13) | 316 (2) | 332 (3) | 318 (8) | A-C; B-C; Ot-City; Up-City; Do-City |
| $Z_m$ 300 m | 308 (5) | 300 (5) | 119 (14) | 180 (2) | 84 (4) | 121 (8) | A-C; A-B; B-C; Do-City; Ot-City; Up-Ot; Up-City; Do-Up; Do-Ot |
| $Z_m$ 100 m | 272 (5) | 228 (5) | 38 (3) | (0) | 16 (1) | 49 (2) | A-C; A-B |
| MBE 300 m | -46 (5) | -78 (4) | -202 (13) | -122 (2) | -256 (3) | -202 (8) | |
| MBE 100 m | -113 (5) | -160 (4) | -266 (3) | (0) | -317 (1) | -241 (2) | |

**Table 3.** As Table 2, but for 4 August case with 15:00-17:00 (instead of 13:00-15:00) as maximum MLH peaks generally later on this day.

| | A | B | C | Upwind | Downwind | Other | Significant Difference |
|---|---|---|---|---|---|---|---|
| **03:00-05:00** | | | | | | | |
| $Z_o$ | 116 (4) | 116 (4) | 131 (3) | (0) | (0) | 131 (3) | A-B; A-C; B-C |
| $Z_m$ 300 m | 205 (5) | 170 (5) | 155 (11) | 233 (1) | 143 (3) | 150 (7) | A-B; A-C; B-C; Do-City; Ot-City; Do-Up; Up-Ot |
| $Z_m$ 100 m | 162 (5) | 139 (5) | 34 (2) | (0) | (0) | 34 (2) | A-B; A-C; B-C |
| MBE 300 m | 76 (4) | 50 (4) | 5 (3) | (0) | (0) | 5 (3) | |
| MBE 100 m | 44 (4) | 24 (4) | -52 (1) | (0) | (0) | -52 (1) | |
| **Growth rate (m h⁻¹)** | | | | | | | |
| $Z_o$ | 379 (5) | 273 (5) | 449 (10) | 366 (5) | 435 (2) | 596 (3) | |
| $Z_m$ 300 m | 551 (5) | 558 (5) | 348 (11) | 371 (5) | 372 (2) | 306 (4) | |
| $Z_m$ 100 m | 548 (5) | 473 (5) | 633 (2) | 633 (2) | (0) | (0) | |
| **15:00-17:00** | | | | | | | |
| $Z_o$ | 3335 (3) | 3210 (2) | 3342 (9) | 3393 (5) | 3525 (1) | 3196 (3) | A-B; B-C; Do-Ot; Ot-Up; Up-City |
| $Z_m$ 300 m | 3622 (5) | 3578 (5) | 3943 (11) | 3890 (6) | 3407 (2) | 4406 (3) | A-C; B-C; Do-Ot; Do-Up; Up-Ot; Up-City; Ot-City |
| $Z_m$ 100 m | 3611 (5) | 3479 (5) | 3528 (2) | 3528 (2) | (0) | (0) | |
| MBE 300 m | 311 (3) | 401 (2) | 684 (9) | 546 (5) | -278 (1) | 1235 (3) | |
| MBE 100 m | 264 (3) | 299 (2) | 185 (2) | 185 (2) | (0) | (0) | |
| **Maximum (m)** | | | | | | | |
| $Z_o$ | 3417 (3) | 3365 (2) | 3575 (9) | 3676 (5) | 3617 (1) | 3392 (3) | |
| $Z_m$ 300 m | 3937 (5) | 3892 (5) | 4496 (11) | 4491 (6) | 3733 (2) | 5014 (3) | |
| $Z_m$ 100 m | 3830 (5) | 3751 (5) | 3700 (2) | 3700 (2) | (0) | (0) | |
| **20:00-22:00** | | | | | | | |
| $Z_o$ | 377 (4) | 420 (4) | 354 (7) | 205 (3) | 520 (3) | 302 (1) | Do-Up; Do-City; Up-City |
| $Z_m$ 300 m | 289 (5) | 194 (5) | 184 (11) | 148 (6) | 233 (3) | 218 (2) | A-B; A-C; Do-Up; Up-City |
| $Z_m$ 100 m | 232 (5) | 149 (5) | 24 (2) | 24 (2) | (0) | (0) | A-B; A-C; B-C |
| MBE 300 m | -138 (4) | -209 (4) | -135 (7) | -74 (3) | -245 (3) | 10 (1) | |
| MBE 100 m | -168 (4) | -222 (4) | -129 (1) | -129 (1) | (0) | (0) | |

## 4.3 Urban plume

To evaluate the representation of an urban plume in the model for the April case, we analyse cross sections along the mean flow through the Berlin centre (Fig. 9a) and perpendicular to the mean flow (Fig. 9b) at 12:00. At this time, the MURK aerosol is being mixed into the ABL with convective updrafts characterised by high aerosol concentrations (possibly linked to the peak in the emissions, Section 2.2), but there is little variability in MLH along or cross wind, suggesting strong thermal forcing in the entire region. There is consistent agreement between $Z_M$ and $Z_O$ at this time with $Z_M$ within ~100 m at most sites (Fig. 9a,b).

By 21:00, an urban plume has developed in the $Z_M$ along-wind cross section (Fig. 10a). Beginning near the start of urban surfaces at 20 km upwind of the city centre, $Z_M$ begins to deepen until about 15 km downwind where it abruptly falls, and this is roughly aligned with the edge of the city and the beginning of non-built land cover (Fig. 10a). At this time, a relatively cold mesoscale airmass was advecting through the Berlin region, bringing a cooler mixed layer upwind which interacts with the warmer air over the urban surface (Fig. 10a). At this time, Berlin was experiencing a canopy-layer urban

heat island intensity of up to 8 K (Fenner et al. 2024, Figure 7). This warmer urban air is advected downwind of the city overtop a cooler shallow stable layer and MMLH identifies this nocturnal surface layer as the mixed layer, as this is where the sharpest aerosol gradient exists. The ALCs indicate an urban plume with similar horizontal scale and higher $Z_O$ downwind. However, the urban-rural difference in MLH is larger in the model (Fig. 10a). $Z_M$ is lower than $Z_O$ generally in Figure 10, with a low bias across most sites (e.g. Figure 5c). We speculate this could be related to weaker aerosol gradients (cf. August case) in the evening, or a cold bias noted when comparing to ERA5 (not shown). Along the cross-wind direction (Fig. 10b) there is a deeper MLH over the city but with symmetry along the cross section as might be expected.

For the August case at 12:00, convection and overturning are apparent, especially in the cross-wind cross section (Fig. 11b). An exceptionally deep mixed layer (~3 km) occurs in the model, in agreement with $Z_O$ in both directions. As noted previously, at 09:00 the mixed-layer growth is delayed over the city (Fig. 8f). This is not unexpected, as the larger thermal mass of the city results in initially larger storage heat fluxes and delayed turbulent sensible heat fluxes, consequently slowing mixed layer growth over the city, which is supported by surface sensible heat flux measurements in the region (Fenner et al., 2024, Figure 8b). Given the low soil moisture in the surrounding rural area during this heat wave case, we also expect more rapid increases in rural turbulent sensible heat fluxes. Evidence of this urban delay is still apparent at 12:00 in both cross sections, with a higher $Z_M$ in rural areas (Fig. 10a, b).

A southwesterly flow at 21:00 (Fig. 12c) creates an urban plume downwind of the city centre with higher $Z_M$ consistent with the ALC measurements. However, there are large differences between some ALC observations which are < 10 km apart (e.g. ~500 m MAVI and HELL; Fig. 12a), and this large spatial variability adds some uncertainty for evaluating $Z_M$. Upwind of the city on the along-wind cross-section, $Z_M$ is ~100 m whereas 10 km downwind of the city it is nearly 800 m. Farther downwind of the city, as in the April case, $Z_M$ variability is linked to a shallow layer beneath the downwind plume of aerosol and warmer city air. Northwest of the city in the cross-wind cross section (Fig. 12b), a shallow stable layer over rural areas outside the city is evident, whereas the southeast transect shows a deeper nocturnal layer with MLHs ~300 m higher.

In summary, on both case days an evening/nighttime urban plume is noted up to 15 km downwind of the city centre. In comparison with Chicago (Cosgrove and Berkelhammer 2019) and Washington DC (Zhang et al. 2011) the plume effect seen here is somewhat smaller in spatial extent however our measure of the plume is based on MLH characteristics rather than near surface air temperature.

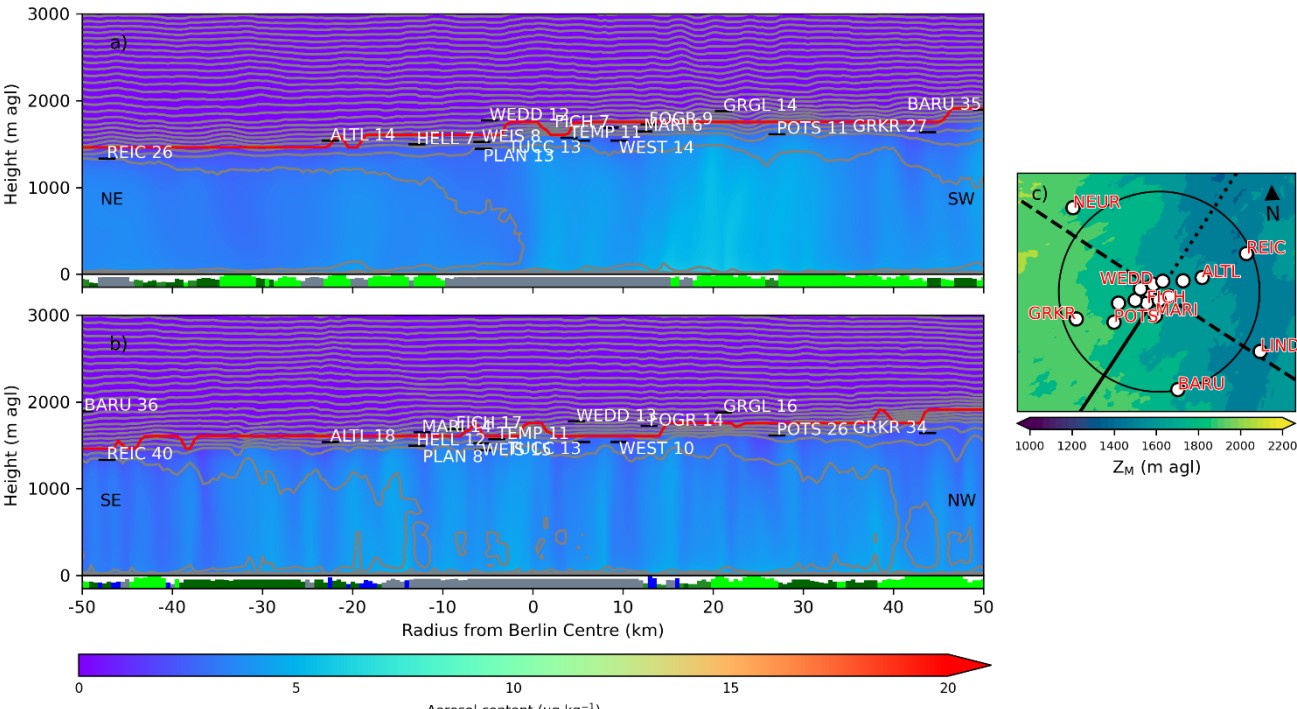

**Figure 9.** (a,b) MURK aerosol content (shaded) for (a) along and (b) cross-wind cross sections through Berlin centre on the 300 m domain at 12:00 UTC on 18 April 2022 with potential temperature (grey 0.5 K isotherms), $Z_M$ (red line) and $Z_O$ (black dashes, site name, distance from cross section [km]). Bottom of (a,b) shows the dominant land cover amount (0-1) along the cross section (grey: urban, light green: C3 grass, green: broadleaf tree, dark green: needleleaf tree, blue: water); and (c) $Z_M$ (shaded), ALC sites, cross section locations (along wind: dotted line=upwind [negative radii], solid line=downwind; cross-wind: dashed line) and 50 km radius (thin black line) from Berlin centre. Cardinal directions are noted at each end of (a,b) to orient cross-sections.

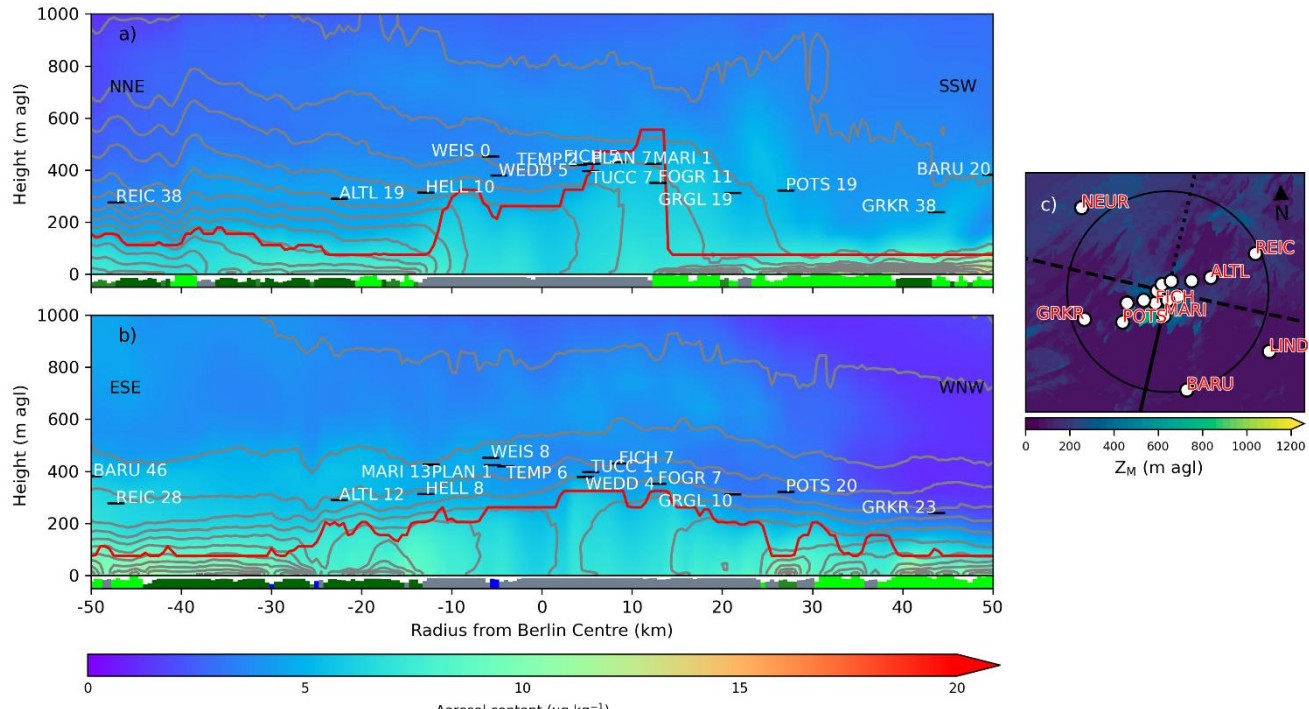

**Figure 10.** As Figure 9 but at 21:00 UTC.

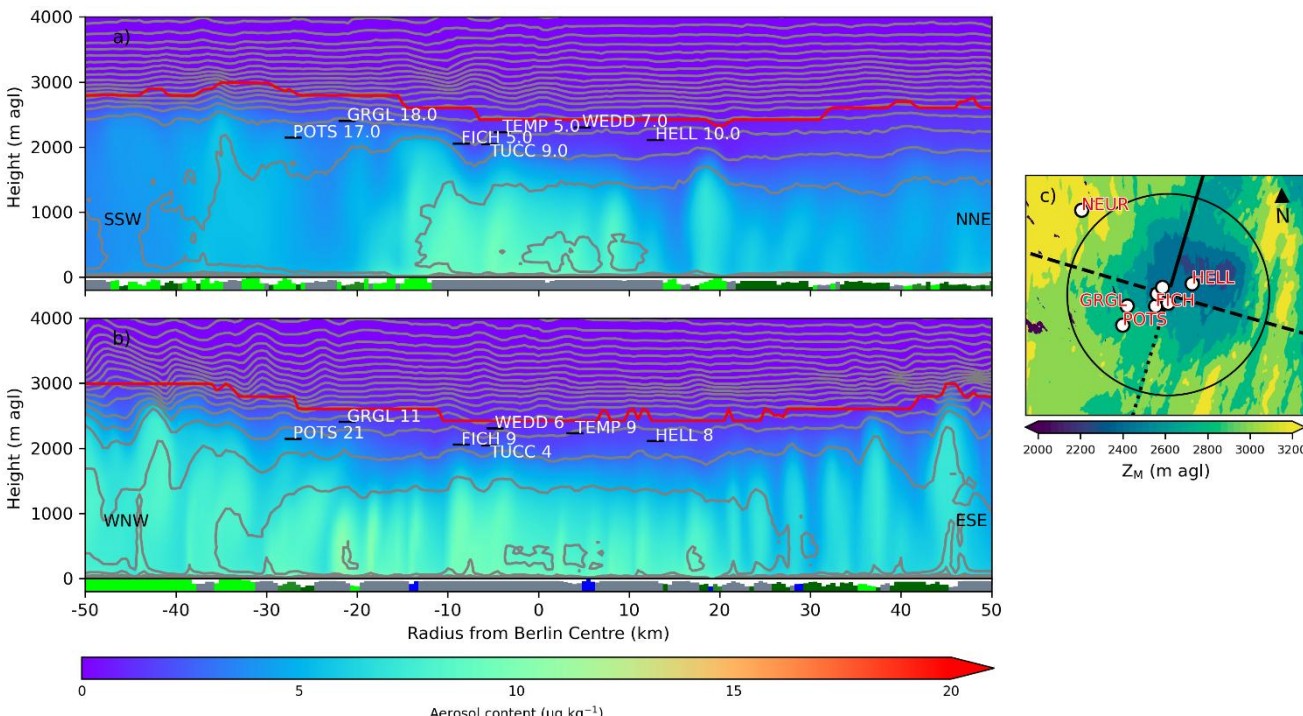

**Figure 11.** As Figure 9, but for 4 August 2022 at 12:00 UTC with the cross-wind transect running from the WNW to ESE.

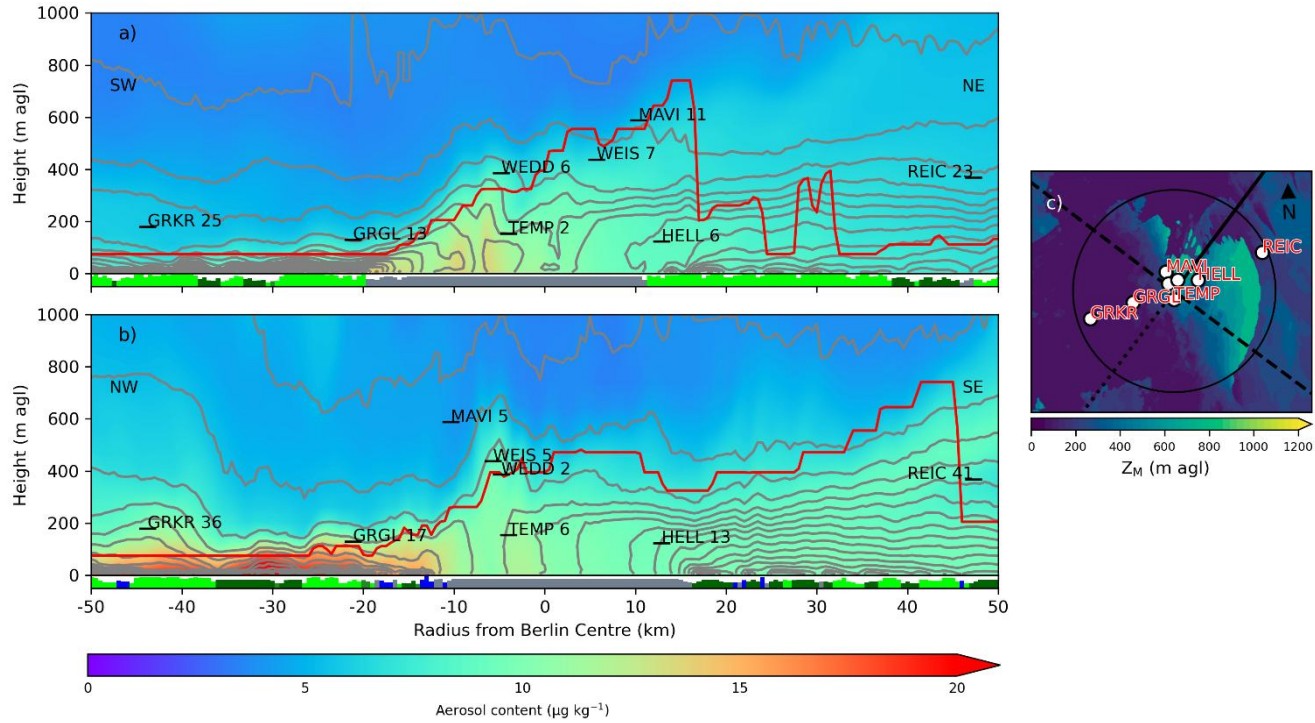

**Figure 12.** As Figure 11 but at 21:00 UTC.

## 5. Summary and Conclusions

In this study, an NWP model with hm-scale grid-spacing is used in conjunction with a dense network of automatic lidar ceilometers (ALCs) from the *urbisphere*-Berlin field campaign to evaluate characteristics and dynamics of the urban mixed layer. Given NWP models at hm-scales are still in developmental stages, this is one of the first tests of their capabilities with such a high density of spatio-temporal mixed layer observations. The *urbisphere*-Berlin campaign provides a rare opportunity for evaluation of hm-models in an urban environment with comprehensive and systematic observations. We develop a novel algorithm (MMLH) which can be applied to Unified Model (UM) MURK aerosol output to obtain fields of mixed-layer height (MLH). MMLH is designed to be comparable with ALC-derived MLH to facilitate evaluation of MLH characteristics and could be used in conjunction with other atmospheric models with aerosol concentration output.

Two days from the *urbisphere*-Berlin campaign (18 April and 4 August 2022) are selected for hm-scale model simulations both of which were characterized by clear-sky, dry convective conditions. In the August case, a heatwave took place in the Berlin region and provided a unique opportunity to evaluate the model under an extreme event impacting an urban

population. The April case was characterized by advection of a cool maritime airmass from the northeast into the Berlin region, with a consistent thermal urban plume evident at night (Fenner et al. 2024).

    Evaluation of MMLH at the ALC sites shows that the maximum extent of the mixed layer is well estimated by MMLH on both days. The 100 m model shows an improved ability, relative to the 300 m model, in discriminating residual layers not associated with the dominant mixed layer (e.g. Fig 4b,d). This is most likely due to a combination of increased vertical

resolution allowing for more finely resolved gradients in aerosol and increased horizontal resolution leading to better representation of turbulence especially during the morning growth phase when it is expected that the 100 m model will begin using the 3D turbulence scheme earlier than the 300 m. Agreement between the ALCs and MMLH is generally better closer to the city centre and decreases towards rural areas outside the city (Fig 5a-b), possibly due to poor representation of suburban land cover in the model.

In the April case, the MLH over the inner city (Ring A) is consistently higher compared to surrounding rural (Ring C) areas in the morning and evening (Fig. 7e,h; Table 2). The spatial extent of the elevated MLH roughly corresponds with the extent of urban land cover in the model simulations (Fig. 7e,h). During the evening, the city's influence on MLH is most obvious in the model.  This is not surprising given the evening corresponds to a period of large thermal disequilibrium between the city, where an upward directed sensible heat flux persists into the night (Fenner et al. 2024), and its rural surroundings

where the sensible heat flux transports energy towards the surface because of more rapid near surface radiative cooling.

    During the August heatwave, the modelled MLH growth is delayed in the city, causing a reduced MLH over the city by ~50% (Fig. 8b,f). This could be explained by a reduced sensible heat flux over the city, which is apparent in observation–model comparison (Fenner et al. 2024). However large modelled latent heat fluxes were also noted in Fenner et al. (2024) and may be indicative of continuing issues with urban soil moisture and thermal initial states in the model. Although the model

initial soil moisture profile is modified (Section S8) to address this extreme drought period, it remains wetter than most observed profiles in Berlin (Figure S9).

    Despite significantly different meteorological conditions, a thermal urban plume is evident in the evening on both days which results in a higher mixed layer up to 10-15 km downwind of the city. Compared to rural areas, in the April case the mixed layer is deeper by ~200 m downwind, whereas on 4 August it is ~500 m deeper. A downwind city plume on 18

April is captured by the ALCs and there is general agreement with MMLH in terms of extent and height of the plume. In the August case there were fewer available ALC observations in the evening and thus the urban plume is sampled with lower station numbers. However, several urban sites match well with the model's urban plume (i.e. MAVI, WEIS Fig. 12a). The deeper vertical extent of the plume in August is consistent with deeper daytime surface and mixed layer heating.

    In this study we have confirmed some physical aspects of the urban plume with hectometric NWP modelling in

conjunction with observations. However, a better understanding of spatial variability within the UBL including if/when a plume occurs and the interaction with neighbourhood and mesoscale turbulence, stability, urban morphology, density and vegetation is necessary. The spatial variability of the plume appears to be closely related to boundary layer stability and thus may have an influence on nocturnal low-level jets (Céspedes et al. 2024). Boundary-layer stability also influences cloud

formation and convective triggering, hence a better understanding of the urban plume may help explain the influence urban
areas have on shallow and deep convection, cloud development and their properties, and resulting convective precipitation
(Forster et al. 2024).

Biases between MMLH and ALC (Tables 2 and 3) are generally two times larger than the difference in vertical
spacing (Fig. S1) between ALC and model levels. The sign of MBE across groups at different times (Tables 2 and 3) is also
very consistent, giving confidence that differences between $Z_O$, and $Z_M$ are not significantly impacted by differences in vertical
spacing. However, at some time periods they are of a similar order (e.g. 03:00-05:00 August case, Table 3). Hence, we cannot
fully rule out the impact of vertical spacing differences in the early morning when the mixed layer can be very shallow.

The new MMLH algorithm developed in this study is an effective tool to diagnose an aerosol derived MLH, distinct
from the thermodynamic ABL depth. MMLH has the advantage of only depending on simple diurnal cycle properties (e.g.
sunset time) and relatively few empirical 'tuning' parameters (e.g. estimate of MLH growth). With the aerosol concentration
profile being MMLH's main input, MMLH should be usable with conventional aerosol schemes used in other NWP models.
Improvements to the MURK scheme (e.g. proposed by Warren et al. (2020)) could improve the representation of dynamic
aerosol processes, which may lead to an improved simulated aerosol profile. In addition, rather than evaluating MLH at one
location (Lean et al. 2019; 2022), MMLH allows evaluation of the full spatial variability of MLH at hm-scale using the UM.
Previously noted discrepancies between MLH, mixing-height, and thermodynamic boundary layer height (Kotthaus and
Grimmond 2018a; Kotthaus et al. 2018; Lean et al. 2022) could be studied with full 4-D model fields instead of individual
profiles.

This study fills an important need for evaluation of the capabilities and deficiencies of hm-models, which are planned
to be implemented operationally over urban domains in the very near future (Hanley and Lean 2024). With recent progress in
the development of hm-models, this study motivates avenues for future research involving the use of more realistic surface
characteristics than currently used in the UM. The CCIv1 land cover has coarse resolution (300 m) and is based on relatively
older data from 2012. Higher-resolution land cover datasets are available, such as the 10 m resolution ESA WorldCover
(Zanaga et al., 2022). However, this remains coarse when resolving individual buildings (Kent et al. 2019). Additionally, there
is opportunity to use more detailed urban morphology data for Berlin (Fenner et al. 2024), instead of relying on empirical
relations developed for London (Bohnenstengel et al. 2011). These data types are available and being processed for the Berlin
region within the *urbisphere* project, for use in future model simulations.

**Acknowledgments.** The authors thank the contributions of the urbisphere team, including Felix Baab, Josefine Brückmann,
Martina Frid, Leonie Grau, Stanley George, Valentina Guacita, Giannis Lantzanakis, Lars Mathes, Emmanouil Panagiotakis,
David Parastatidis, Karthik Reddy Buchireddy Sri, Dirk Redepenning, Sebastian Scholz, Ingo Suchland, Timothy Sung,
Angela Wendnagel-Beck, Fred Meier, Matthias Zeeman, Jörn Birkmann, Nektarios Chrysoulakis, Matthew Clements, Denise
Hertwig, Kai König, Zina Mitraka, Dimitris Poursanidis, Dimitris Tsirantonakis, Carlotta Gertsen, Jonas Kittner, Yiqing Liu,
Matthew Paskin, Beth Saunders and Yuting Wu. This work is funded by the European Research Council (ERC) under the
European Union's Horizon 2020 research and innovation program (Grant Agreement 855005), UKRI NERC ASSURE
(NE/W002965/1) and UKRI EPSRC FUTURE (EP/V010166/1). We also thank the urbisphere partners who operated and

provided sites for ALC installations, including Deutscher Wetterdienst, Technische Universität Berlin, and Freie Universität Berlin. Computing resources for model simulations were provided by Monsoon2, a collaborative facility supplied under the Joint Weather and Climate Research Programme, a strategic partnership between the Met Office and the Natural Environment Research Council.

**Author Contribution. RHG:** Conceptualization; Formal analysis; Investigation; Methodology; Software; Visualization; Writing – original draft preparation; Writing – review & editing; **SG:** Conceptualization; Funding acquisition; Methodology; Writing – review & editing; **LB:** Conceptualization; Data curation; Methodology; Software; Writing – review & editing; **DF:** Conceptualization; Data curation; Methodology; Writing – review & editing; **HL:** Conceptualization; Methodology; Writing – review & editing; **AC:** Conceptualization; Methodology; Writing – review & editing; **WM:** Data curation; Writing – review & editing; **DL:** Data curation; Writing – review & editing; **JKPS:** Software; Writing – review & editing

**Competing interests.** The authors declare that they have no conflict of interest.

**Data availability.** Related data and code used in this work will be placed in an appropriate citable repository on Zenodo.

**Appendix A.**

*A1 Normalized mean absolute error*

The mean absolute error (MAE) between $Z_M$ and $Z_O$ is defined as:

$$MAE(Z_M, Z_O) = \sum_{i=1}^{n} \frac{|Z_M(i) - Z_O(i)|}{n} \tag{1}$$

where $n$ is the number of observations. We define the normalized MAE (nMAE) as:

$$nMAE(Z_M, Z_O) = \frac{MAE(Z_M, Z_O)}{mean(Z_M)} \tag{2}$$

*A2 Hit Rate*

In defining the 'hit rate' (HR), a 'hit' is counted at a given site when $Z_M$ is within a given window of the $Z_O$ value:

$$[Z_O + (Z_O * w)] > Z_M > [Z_O - (Z_O * w)] \tag{3}$$

For the $\pm 10\%$ and $\pm 50\%$ windows $w$ is 0.1 and 0.5 respectively. A 'hit window' ($\Delta H$) is then defined as the range in which a hit can occur, i.e. $\Delta H = [Z_O + (Z_O * w)] - [Z_O - (Z_O * w)]$. Counting the number of hits for each window ($h_{10,50}$) at a given time we define the $\pm 10\%$ and $\pm 50\%$ 'hit rate' ($HR_{10}, HR_{50}$; in %) for a given time as:

$$HR_{10} = \frac{h_{10}}{N} * 100, \qquad HR_{50} = \frac{h_{50}}{N} * 100, \tag{4}$$

where $N$ is the total number of ALCs giving data at the time. We define a normalised hit rate for the 10 and 50 % windows (nHR$_{10,50}$) by dividing the number of hits by the hit window,

$$nHR_{10} = \frac{h_{10}}{\Delta H} * \frac{1}{N} * 100, \qquad nHR_{50} = \frac{h_{50}}{\Delta H} * \frac{1}{N} * 100. \tag{5}$$

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
