# Peer review of "Hectometric-scale modelling of the mixed layer in an urban region evaluated with a dense LiDAR-ceilometer network"

_EGUsphere, 2025_

## Author Comment (AC1)

We thank the Reviewer for their comments. In the following we give :

 *Black -reviewers comments*

 Purple - Our response

**Reviewer 1**

*This manuscript aims to evaluate hectometric-scale modeling of the urban mixing layer using an extensive ceilometer network. Specifically, it seeks to assess the model's ability to determine the urban ABL, focusing on the mixing layer height (MLH), by comparing simulated aerosol vertical profiles with ceilometer measurements using the same retrieval algorithm. The model's approach to determining ABL/MLH turbulent mixing and its relationship to aerosol vertical distribution is not adequately discussed. Therefore, it remains uncertain whether this evaluation is truly useful in evaluating the urban mixed layer. The study then assesses the model's ability to capture the spatial distribution of MLH and evaluates the urban smoke plume for two case study days. These results suggest that the model still does not accurately represent land characteristics, which is an important and interesting finding, supported by previous research. Additionally, the impact of this misrepresentation on urban plume transport is an important result, and further insights on this should be included. The manuscript would benefit from a more thorough discussion of the implications of these findings, explaining how these models are to be implemented and how the paper's findings may influence that implementation. Overall, the reviewer recommends publication after addressing major revisions.*

**Major Comments:**

1) *The manuscript demonstrates an inconsistent use of terminology regarding the mixing layer height (MLH) and its relationship to the turbulent or thermodynamic ABL and aerosol distribution. While acknowledging that aerosol-based MLH may differ from thermodynamic MLH, the manuscript frequently equates them or uses them interchangeably. It is crucial to consistently differentiate between these terms.*

While acknowledging there are three terms defining ABL height which are discussed in the Introduction, the authors are unsure at which points later in the manuscript there is inconsistency in the use of these terms. We have made changes at **L85-90** to more clearly define the differences between 'mixing height', 'mixed layer height', and thermodynamic ABL height with an additional reference added to Kotthaus et al. (2023a) which clearly discusses the differences in turbulence, aerosol, and thermodynamically based metrics for ABL height.

Removed the sentence **(@L75 of original manuscript**) to remove potential confusion regarding terminology.

**L71-72:** MLH is defined and is the only term used to refer to the aerosol based vertical extent of recent mixing throughout the manuscript.

As in **Figure 4** we show a thermodynamically based metric for ABL height ('BLD diagnostic') we clarify the text (**at L245**) - that this is distinct from MLH. These results are intended to highlight the difference between them in the late afternoon on 4 August, and discuss previous level of agreement (e.g. Kotthaus et al. 2018; Kotthaus et al. 2023a). ALC based MLH and thermodynamically derived ABL height is a focus of this paper. No other new metrics or terminology are introduced, and we consistently use the 'mixed-layer' or 'mixed layer height' terms, as this is the focus of this study.

2) *Similarly, the authors should carefully consider the distinction between "mixed" (e.g. L70) and*

*"mixing" (e.g. L84) layers. The term "mixed layer" hints at a final, fully-mixed state, while the "mixing layer" generally describes the convective daytime ABL in the context of continuous mixing. Ensure clear and consistent usage of these terms throughout the manuscript.*

See also our response**: Major Comment 1**

Paragraph clarified (**L85-90**) as it is essential these terms are clearly distinguished.

We clarified the MLH definition (**L71-72**) ' the vertical extent of recent mixing' - i.e. the MLH involves turbulent processes which could be ongoing.

Following previous literature (e.g. Barlow et al. 2011, Scarino et al. 2014; de Bruine et al. 2017; Bonin et al. 2018; Kotthaus et al. 2018; Lean et al. 2022; Fenner et al. 2024) we distinguish between mixed layer height (MLH) and mixing height (MH). These are defined differently because of the atmospheric variables consider differ (expanded **L85-91**).

Earlier (**L38-43**) we addressed which measures we use for defining the ABL height.

3) *The term "aerosol layer height" may be more accurate than "mixed layer height" in this context, especially considering that, as shown in Figure 4 for instance (afternoon decay period), turbulence is no longer the dominant driver in the model. This suggests that the term "mixed layer" may not be appropriate when discussing the ZM or ceilometer-based ZO retrievals. When addressing the links to a thermodynamic ABL, then using the BLD would be more appropriate.*

Similarly with the studies listed in the response to **Major Comments 1 and 2**, there is a significant amount of literature using the term 'mixed layer height' based on aerosol measurements from ALCs (Scarino et al. 2014; de Bruine et al. 2017; Kotthaus et al. 2018; Lean et al. 2022; Fenner et al. 2024). We follow this terminology.

As our aim is to evaluate the model with ALC derived MLH, we develop MMLH (Section 3) following on from earlier forward operators (e.g. Warren et al. 2018; 2020) to ensure the model is comparing consistent variables to that observed (**L89-90**). In this study, the MLH is obtained from aerosol gradients  to identify the 'mixed layer' from both MMLH-model and ALC-sensors, following previous literature.

As noted in the response to **Major Comment 1**, the Unified Model BLD diagnostic is used to illustrate the difference (Fig. 4) between an aerosol-based and a thermodynamic based metrics. Our goal is not to assess which is better/poorer but to highlight their difference – essential for all model evaluations and data assimilation that consistent model and observations variables are used.

4) *The manuscript mentions the use of a turbulence scheme but does not explain how it influences the simulation of ABL nor how a BLD is estimated. It is important to clarify how the model defines the vertical extent of surface aerosols and how the turbulence scheme interacts with this vertical distribution in order to better assess and evaluate the model, which is a stated objective of this study.*

Discussion has been added on the influence of the turbulence scheme on the simulations in the 100 and 300 m model (**L133-136**) and in the conclusions **(L423-424)**.

Essentially, the influence of the turbulence scheme is through its partitioning between 3D and parameterized turbulence. In the 100 m domain a greater portion of the turbulence in the ABL will

be resolved by the 3D scheme and will happen earlier in the day (cf. 300 m). This is important for our results regarding the growth period in the morning and why the 100 m domain shows improvement over the 300 m.

Regarding the BLD diagnostic, the parcel method defines the BLD as the height to which an air parcel can rise dry adiabatically from the surface until it intersects with the environmental temperature profile (Holzworth 1964). Text added at **L245-248**.

The aerosol emissions are vertically distributed near the surface in the model, for most emissions (i.e. area sources) they are spread out across model vertical levels from the surface up to 150 m. For point sources, there is a distinction made for large and small sources, of which small sources are treated in the same way as area sources. For large point sources, emissions are spread evenly over model levels within a known minimum and maximum plume height for the source. If the plume heights are not known the minimum is set to 150 m and the maximum set to 365 m. Explanation added: **L147-150**.

5) *The manuscript lacks a detailed explanation of the aerosol prediction methods and aerosol scheme. A more thorough description of these processes is essential for a comprehensive assessment of the model's capabilities in simulating the urban boundary layer and urban plume.*

More thorough description of the aerosol scheme now given: **L142-146**.

The aerosol scheme treats the pollutants as a single-aerosol species, which is a prognostic quantity that is advected and mixed in the model consistent with other scalars. Removal of aerosol is accounted for via precipitation (although not relevant for these cases) and non-emitted sources via conversion factors.

6) *The manuscript uses both the CABAM and STRATfinder algorithms for ALC retrievals. While both algorithms aim to retrieve the ZO, the differences between them and the implications of using two separate algorithms should be discussed. It is important to address how these differences may affect the comparison to the ZM, as this could influence the results and interpretation.*

Kotthaus et al. (2020) note these MLH algorithms are tailored to specific ALC types for implementation across diverse ALC sensor network (i.e. as *urbisphere*-Berlin had). From Kotthaus et al.'s (2020) detailed analyses of differences, they conclude there is very good agreement in MLH from high (e.g. CL61, CHM15k - STRATfinder) and lower SNR ALC (e.g. CL31 - CABAM) in diverse ABL conditions. The largest discrepancy occurs in the evening and early morning when detecting shallow layers due to the ALC differences in SNR and optical overlap. Additionally, STRATfinder has a delayed decline in the mixed layer around sunset (cf. CABAM), likely due to challenges of tracking very subtle gradients in attenuated backscatter as a shallow nocturnal layer forms. Discussion added **(Section 2.3) L180-187**.

7) *Section 4 would benefit from specific quantitative values when discussing over- and underestimations. This would provide a clearer understanding of the magnitude of these errors and allow for a more precise evaluation of the model's performance.*

**Added Tables 2 and 3 to supplement analysis of Fig. 7 and 8 along with text and appropriate references in Sections 4.2 and 5.** The tables give metrics for various station groupings used to evaluate $Z_M$ using the $Z_O$ data. These spatial groupings are used to analyse the maximum MLH and

growth rates, following the same methodology as Kotthaus and Grimmond (2018b). The analysis is much improved with these results.

[revised manuscript text omitted]

8) *How much of the bias can be attributed to differences in vertical resolution between ceilometers and models? This should be addressed earlier in the manuscript to assess the extent of the biases presented.*

The data processing of the ALC backscatter for the STRATfinder algorithm yield a constant vertical spacing of 30 m up to ~15km height. For CABAM the vertical spacing is maintained for the original CL31 range gates which is 10 m up to ~7.5 km height. This has been added to **L186-187**. Comparing these to the model levels of our simulations is shown in the figure below. '0p3_L70' is notation for the 300 m domain with 70 vertical levels, and '0p1_L140' is the 100 m domain with 140 vertical levels. A description of the model vertical spacing has been added at **L123-124**. The left figure shows the vertical spacing in the lowest 1000 m along with the minimum MLH found among the ALC sites for ZO, and ZM. The lowest MLH diagnosed from any of the ALC data was 68 m, and for the model this was 75 and 16 m for the 300 m and 100 m domains, respectively. At these very low heights (< 200 m) the vertical spacing between the ALC and model is comparable. In terms of the biases found, if we look at Table 2 and 3 for the results during early morning (03:00-05:00) when we might expect these lowest values to occur, the MBE are generally of an order larger than any difference in vertical spacing. However, the MBE is somewhat smaller for the August case (Table 3), and the difference between CABAM and the 300 m domain in terms of vertical spacing is similar to the MBE.

In the right figure, the 99th percentile of MLHs from the ALC sites is shown as a horizontal line for ZO, and ZM. For the ALC measurements this was nearly 3700 m (from the August case), 4800 m from the 300 m domain, and 4100 m from the 100 m domain. At these heights the 100 m and ALC data have a difference in vertical spacing of around 100 m, and for the 300 m domain this is ~250 m. Looking at the MBEs for the August case for afternoon (15:00-17:00; Table 3) the differences between ZO, and ZM are generally larger than these differences in vertical spacing and when looking at the differences in the 'Maximum' in Table 3, the differences between ZO, and ZM are again larger than differences in vertical spacing. The sign of MBE across groups at different times in Tables 2 and 3 is very consistent, giving confidence that there is a clear result between ZO, and ZM that is not significantly impacted by differences in vertical spacing. However, because the differences in vertical spacing and MBE are at times of the same order (e.g. 03:00-05:00 August case) we cannot rule out an impact from this. This discussion has been added in **Section 5 L454-458** and the figure below has been added in **Supplemental Material Figure S7** and referenced in the discussion in **Section 5**.

Figure S7 A comparison of the vertical spacing of data used for MLH in this study for (left) the first 1000 m in CABAM (dotted blue), STRATfinder (dotted red), the 300 m domain ('0p3_L70', solid dark blue), and the 100 m domain ('0p1_L140'; dashed dark blue). Brown lines indicate the minimum MLH detected at any site by MMLH ($Z_M$) in the 300 m (dashed brown), and 100 m domains (dotted brown), and for all ALC observations ($Z_O$, solid brown). (Right) Vertical spacing with height up to 6000 m with brown lines indicating the same as in the left but now showing the height of the 99th percentile of MLH among all

[Figure]

sites.

9) *Section 5 should be revised in light of the points above. If the authors can better support the rationale for using simulated aerosols for MLH, and ensure that the definitions of thermodynamic and aerosol-based layers are consistent, the manuscript will be clearer and more coherent.*

We have revised the manuscript in light of these points regarding the responses to **Major Comments 1, 2, and 3**.

10) *The manuscript's title, 'Hectometric-scale modelling of the urban mixed layer evaluated with a dense LiDAR-ceilometer network,' suggests a primary focus on the urban mixed layer. However, a substantial part of the manuscript is dedicated to evaluating the urban plume. A revision of the manuscript's stated goals to ensure better alignment between the title and the content is needed.*

Our stated focuses include both the urban plume (**L101**) and to evaluate the variability of MLH between urban and rural areas (**L100**), both are concerned with the MLH in an urban region. We modify the title *'Hectometric-scale modelling of the mixed layer in an urban region evaluated with a dense LiDAR-ceilometer network'* to better reflect these goals. Clarification of goals added in the abstract **L16-17**.

11) *Furthermore, if a central goal is to investigate the simulation of the plume, it is crucial for the authors to explain why the analysis predominantly focuses on the top of the aerosol layer (identified as ZM/ZO) rather than providing an examination of the full vertical and spatial distribution of aerosols. A detailed analysis of the entire plume's structure would offer a more complete understanding of the model's performance in simulating urban aerosol transport.*

There are two main stated goals of the paper which we address in the response to **Major comment 10**. The algorithms used are designed to identify the aerosol layers associated with a MLH. During growth conditions there is typically an abrupt decrease in aerosol concentration above the mixed layer. At night multiple aerosol layers may be present, and the algorithms are designed to identify the layer most likely associated with the MLH. We analyse the manifestation of the urban plume through our MLH observations and simulations, but our focus is not on the detailed plume structure (e.g. thermodynamic profile). However, Fig. 9-12 do present qualitative analysis of the thermodynamic and aerosol structure of the plume. Our future work will involve more detailed analyses on urban plume controls, structure, and thermodynamic profile stability.

12) *The misrepresentation of soil moisture in the model is repeated throughout the manuscript, it would be helpful to see the extent of this misrepresentation and how it may relate to the findings in the study.*

Supplemental S2 gives the methods used to adjust the bias in soil moisture over urban areas for the simulation. Figure S3 presents observed soil moisture profiles in comparison with the model at an urban site to demonstrate that the adjusted soil moisture profile is in general agreement with other observed profiles around Berlin.

**Minor Comments:**

- *There is a discrepancy between the text and figure captions for Figures 4, 6, 7, and 8, where it appears that the images for 300m and 100m simulations were swapped.*

  Fixed

- *Figures 7, 8, 9, 10, 11, and 12 are difficult to interpret. Increasing their size and resolution, especially for the ALC circle markers, would improve clarity, as it is currently hard to see the bias shading.*

  To improve readability the size of the labels in Fig. 9-12 are increased, as are marker sizes in Fig. 7 and 8 (bottom rows).

- *Section 2.1 would benefit from a more detailed description of the region under study.*

  It was felt that this detail could be more suited for Section 2.3 and have added this at lines **L177-179**.

- *Regarding Figure 1, was the synoptic setup validated using observational data or reanalysis?*

Comparing **Fig. R1** (ERA5) and Fig. 1 we can see April (left) wind flows are very similar, but the cooler airmass to the northeast which is advecting into the Berlin region is delayed and less robust (cf. global UM). This results in a warmer airmass (~2 C° at 850 hPa) in the Berlin region in the afternoon, which is not expected to significantly impact our MLH assessment. The regional flow regime is more important to capture. In August (right) we see a very similar setup (cf. Fig. 1b), with the advection of a warm airmass from the southwest captured.

Fig. R1: As Figure 1, but using ERA5 (Hersbach et al. 2020) for (left) April and (right) August cases.

[Figure]

- *The caption for Figure 2 is unclear and should be revised for clarity.*
  Updated

**Figure 2.** Berlin study area with (a) the extent of the three nested model domain (brown, 100 m, 300 m, and 1.5 km) and rings B and C (black circles) around Berlin (red star), and (b) location of the 25 ALC sites (blue, with WEDD [Wedding] - yellow star) with the 300 m domain model dominant land-cover (colour, BL = Broadleaf, NL = Needleleaf) and orography (30 m contours), plus the A to C rings (black circles) at 6, 18 and 90 km radius from Berlin centre, locating Inner city, Outer city and rural areas, respectively (Fenner et al. 2024) and extent of 100 m model domain (brown box).

- *It would be helpful to label parts A, B, and C in Figure 2 for better clarity.*
  In Figure 2b the labels for the rings A, B and C are now **bold** to improve clarity.

[Figure]

- *L168 requires a reference for the statement. The claim that CL31 and CL51 sensors "reach sufficient overlap at lower heights" is questionable, as these systems typically achieve overlap at approximately 100m. These sensors also have known artifacts and signal issues, as noted by*

*Kotthaus et al. (2016) - "Recommendations for processing atmospheric attenuated backscatter profiles from Vaisala CL31 ceilometers".*

From Kotthaus et al. (2020): 'The CL31 reaches complete optical overlap at ~70 m range; however, corrections for near-range artefacts are necessary.' These corrections for near-range artefacts are developed in Kotthaus et al. (2016) and Kotthaus and Grimmond (2018a) and were applied in this study. For CL61 the overlap (from PROBE https://zenodo.org/records/11211873) is 35 m.

**L191** is meant to draw comparison between the CHM15k and CL31/61, and to state that the CHM15k optical overlap is incomplete <200 m (cf. CL31/61 which is ~70 m).

Reference (Kotthaus et al. 2020) added (**L192**) to support this.

- *L214: Should this refer to Figure 4b?*
  Yes, done.
- *L221: how is the BLD defined or simulated?*

  Addressed in response **Major Comment 4**. Description of the BLD is added **(L245-248).**

- *For Figure 5, it would be useful to describe the image in order of a, b, c, and d.*
  Done.
- *In L281-282, the references to Figures 7c and 7g are unclear regarding the 'shallow downwind' and 'high upwind' values. The visibility of markers in these figures is poor, making it difficult to discern the discussed trends. To improve clarity, it would be beneficial to include specific quantitative values in the text to support these observations.*

[Figure]

Fig. 7 and 8 markers for the 100 m data are enlarged, but not for 300 m as some stations would be unreadable.

Quantitative analysis added (Tables 2 and 3) and addressed in **Major Comment 7.**

References not in text:

de Bruine, M., Apituley, A., Donovan, D. P., Klein Baltink, H., and de Haij, M. J.: Pathfinder: applying graph theory to consistent tracking of daytime mixed layer height with backscatter lidar, Atmos. Meas. Tech., 10, 1893–1909, https://doi.org/10.5194/amt-10-1893-2017, 2017.

Hersbach H, Bell B, Berrisford P, et al. The ERA5 global reanalysis. Q J R Meteorol Soc.; 146: 1999–2049. https://doi.org/10.1002/qj.3803, 2020.

---

## Author Comment (AC2)

We thank the Reviewer for their comments. In the following we give:

      *Black -reviewers comments*

      Purple - Our response

**Reviewer 2**

*The authors address the very relevant topic of the recent development of atmosperic numerical models towards the hectometric scale. The continuous increase of computing power meanwhile allows several Meteorological Services to run their mesoscale models at an hectometric scale, or they are close to it. One obvious application at that scale is to simulate the weather and climate phenomena over urban areas. In the recent years, several Weather Services have developed and integrated urban schemes in their NWP models.*

*A remaining issue for activating these urban schemes in operational NWP is the availability of observational datasets covering rural and in particular urban areas. The authors present and utilize a dense LiDAR-ceilometer network over Berlin.*

*General comments:*

*1) The strategy in this manuscript is focussed on the MURK aerosol scheme in the UM. Could this method also be transferred to a sophisticated aerosol model, if becoming available in UM, or to other atmospheric models in general. There is one sentence like this in section 5, maybe this could be a bit elaborated?*

This method could be transferred to a more complex aerosol scheme. Warren et al. (2020) proposed improvements for the MURK scheme to better represent dynamic aerosol processes. More sophisticated aerosols models would likely have improved dynamic aerosol processes, which would be beneficial for a wider range of meteorological conditions.  The main input for our algorithm is the aerosol concentration profile which should be available as output from a conventional aerosol scheme. Text added **L461-462**.

A large dependence however is also on the NWP model configuration itself, as the aerosol profile is determined through the dispersion of the aerosol as a passive scalar. An important advantage here is the hm-scale model configuration which allows for explicit turbulence with a high vertical resolution in the mixed layer, compared with for example, kilometre-scale models.

*2) Likely also MURK has errors or shortcomings. How much do they influence or limit the general findings of this study?*

We have added more detail about MURK and how emissions are dispersed from the surface in the model (**L142-150**; also see **Reviewer 1 response #4**).

MURK scheme intentionally uses simple conversion to aerosol (i.e. aerosol sources via non-emissions) and as previously mentioned Warren et al. (2020) proposes several improvements to the scheme (added at **L463**). For this study, we would expect this impact to be minimal given the clear, dry conditions and short simulation period. Another possible shortcoming is the resolution of the emissions dataset which needs to be interpolated down to a 100 m grid from the 0.1° data. Because of this we lose intracity variability in the emissions which may then translate to a loss in variability in the aerosol profile on the intracity scale. We do expect this to be compensated by the model's resolution of intracity variability however.

*3) I find Fig. 4 very convincing. But it stays a bit unclear, why the UM boundary layer diagnostic BLD is so very different from the aerosol-based method, particularly in the afternoon and evening io 4 August. Some literature is quoted confirming this result. Why does the parcel-based BLD not capture the effect that the aerosols are actually further transported verically upward? By which*

*physical process does this happen? Maybe a figure of the vertical motion (in a cross section) could shed more light on this?*

This is an interesting point. We add Fig. S8 with the environmental lapse rate (averaged to 15-minutes) and Fig. S9 with the vertical velocity (15-minute instantaneous) to provide some explanation.

The BLD diagnostic on 4 Aug (Fig S8b, d) strictly follows a weak inversion throughout the afternoon and evening. Whereas, the majority of strong overturning is below the BLD in the afternoon (Fig. S9b,d). In the evening there is still strong vertical motion reaching above the BLD. Thus, some additional mixing is possible above the weak thermal inversion and may lead to the rise in $Z_M$ even in the late afternoon until sunset. In similar hectometric simulations, Lean et al. (2022) found wave features above the mixed layer which may provide some explanation for what is seen in Fig. S9. Text added at **L251-254.**

[Figure]

Figure S8. As Fig. 4, but showing the modelled environmental lapse rate per 100 m averaged to 15-minutes (colour bar), with BLD diagnostic (grey lines), $Z_M$ (black lines), and sunrise and sunset (dashed lines).

[Figure]

Figure S9. As in Fig. S8 but showing the 15-minute instantaneous vertical velocity (W [m s$^{-1}$]) .

*4) Section 4.2 has a very descriptive character. Would it possible to also find some reasoning or explanations for the things described?*

See also **Reviewer 1 response 7**. We have added Table 2 and 3 along with text in Section 4.2 which offer quantitative results to go along with what is seen in Fig. 7 and 8. We believe the results in Section 4.2 are clearer when grouping stations by rings (i.e. urban-rural variability) or by upwind/downwind of the city.

*5) Figs. 9-12 look very convincing. Anyway, are there ideas about the (remaining) differences between model and obs in Figs.10 and 12?*

The April case appears to have a low bias in the evening (Fig. 10), which we attribute to the difficultly with weaker and finer gradients in the evening. Another possible explanation is the cold bias noted when comparing to ERA5 (see **Reviewer 1 Minor comment #4**) which may have led to a shallower mixed layer in our simulations, compared to the ceilometers.

*Minor comments:*
*- Fig. 1: The rings are not explained well. The explanation comes in the caption of Fig. 2. Maybe move this to caption of Fig. 1, or refer to Fig. 2?*
Fig. 1 caption updated to explain this.

*- Fig. 2: The descriptions of rings A to C are not so easy to find. Maybe different colour?*
Fig. 2 rings labels enlarged and made bold.

*- L. 172: ... observed MLH (from now on referred to as Z_O) ...*

modified to (hereafter $Z_O$)

- *L. 174: Similar with model MLH Z_M.*

modified to (hereafter $Z_M$)

- *L. 209: WEDD: Maybe add the name „Wedding" of the quarter (for the readers who know Berlin 😊).*

'[Wedding]' added to Fig. 2 caption. Incorrect reference to 'Fig. 1b' at **L233** corrected to Fig. 2b.

---

## Author Response (AR2)

*1) lines 86f and 92f: In this new paragraph, you provide more detail on different metrics for ABL height detection. 'MLH can be derived from analysing a concentration profile (e.g. aerosol concentration, temperature, moisture)…' and '… concentration profiles of temperature, moisture, and aerosols…'. Temperature is not a concentration. Please correct and revise.*

Updated.

*2) What does MURK stand for? Please explain.*

MURK refers to one of the original goals in it's development in the operational Unified Model which was visibility prediction. MURK does not specifically stand for a longer term. In Clark et al. 2008 the scheme was not given a name but has since been referred to in later papers as 'MURK' (e.g. Lean et al. 2022). We have modified L92 in order to make this clearer.

*3) Line 123f: '.. vertical spacing below 200 m is ~60 m increasing to ~120 m…'. This implies that the vertical spacing below 200 m is ~60 m at all heights, which it is clearly not based on Fig. S7. Perhaps say something like '.. less than ~60 m below 200 m increasing to …'*

Updated.

*4) Supplemental figures: The order of the supplemental figures needs to be modified to match the occurrence in the main text. For example, Fig. S7 should be Fig. S1.*

Updated this.

*5) Line 149-150: Information on the minimum and maximum plume height are given in the response to the reviewer's comments, but not in the main text. Please include this relevant information in the text as well.*

Added this information at L150.

*6) Line 191: Reference to Fig. 6 is not in order. Figures need to appear in the order they are referenced.*

Reference removed.

*7) Line 245ff: Thank you for including information on the BLD diagnostic, including how it is determined under unstable conditions. How is BLD defined under stable conditions? It is shown in Fig. 4 during the night, so this information is relevant.*

In stable conditions it is the first height where the gradient Richardson number becomes super critical (>0.25). This has been added at L250.

*8) Line 251ff: Evidence of enhanced vertical mixing is not clearly visible in the 300 m run on August 4 during the time of the discrepancy in BLD and MLH (Fig. S9 b,d), raising some doubt in the explanation given in the text. Could horizontal advection play a role? Did you investigate this?*

It is quite possible that horizontally advected aerosol is playing a role; however, we have not focused on this in the analysis. The enhancement of vertical mixing in the Aug. case compared to the Apr. case is notable and was worth describing; however, further qualification is needed to make clear that this is speculative in explaining the difference. L254 has been modified.

*9) Sect. 4.2: This section benefits from adding the quantitative tables as requested by reviewer 1, but is still fairly descriptive. Please consider adding some explanations of the findings as requested by Reviewer 2 (comment 4).*

We have modified the text at L310-313, L326-331, and L340-341 in order to make descriptions clearer and add physical explanations.

*10) Sect. 4.3: Reviewer 2 (comment 5) raises the question about the differences between the model and the observations in the evening. For example, there are quite substantial differences in MLH upwind and downwind of the urban area in Fig. 10 and 12. Please add additional explanation to the text, to address this comment.*

We have added additional explanation to the text at L375-377, and L389 in order to discuss the difference between model and observations.

---

## Author Response (AR3)

*In l. 88, it still says 'concentration profiles of temperature'. Please correct.*

We have removed 'concentration' from this line.

-A minor correction was made to Fig 11a so that distance labels were rounded to whole numbers as in Fig 11b.

-The 'Data Availability' section has been updated in the manuscript with links to Zenodo DOIs where the reader can find data and scripts used in the paper.